# Tuning parameters for polygenic risk score methods using GWAS summary statistics from training data

Wei Jiang [1], Ling Chen[2], Matthew J. Girgenti [3] & Hongyu Zhao [1] ✉

Various polygenic risk scores (PRS) methods have been proposed to combine the estimated effects of single nucleotide polymorphisms (SNPs) to predict genetic risks for common diseases, using data collected from genome-wide association studies (GWAS). Some methods require external individual-level GWAS dataset for parameter tuning, posing privacy and security-related concerns. Leaving out partial data for parameter tuning can also reduce model prediction accuracy. In this article, we propose PRStuning, a method that tunes parameters for different PRS methods using GWAS summary statistics from the training data. PRStuning predicts the PRS performance with different parameters, and then selects the best-performing parameters. Because directly using training data effects tends to overestimate the performance in the testing data, we adopt an empirical Bayes approach to shrinking the predicted performance in accordance with the genetic architecture of the disease. Extensive simulations and real data applications demonstrate PRStuning's accuracy across PRS methods and parameters.

The advent of genome-wide association studies (GWAS) has led to the discovery of numerous loci associated with the most common diseases[1]. These discoveries also provide the opportunity for predicting risks from an individual's genotypes[2]. Accurate genetic risk prediction can enable us to identify high-risk individuals and facilitate disease prevention and early treatment[3].

Polygenic risk score (PRS) is commonly used in genetic risk prediction due to its simplicity and resulting from the additive assumption. Both empirical and theoretical studies have shown that the additive component is expected to account for most of the genetic variance of complex traits[4]. Based on this additive assumption, PRS sums the allele dosages of single nucleotide polymorphisms (SNPs) weighted by their estimated effect sizes[5].

Various PRS methods have been proposed to estimate the effect sizes of SNPs from a GWAS dataset. Compared to individual-level genotype data, summary statistics are more accessible without security and privacy concerns[6,7]. Many PRS methods proposed recently estimate SNP effects with GWAS summary statistics. One of the simplest is clumping and thresholding (C+T)[8-14], in which linkage disequilibrium (LD) clumping is applied to the SNPs that pass a $p$-value threshold. Another related method is pruning and thresholding (P+T), which only includes the SNPs whose $p$-values exceed a threshold after LD pruning. Both LD clumping and LD pruning are step-wise heuristic procedures that select a set of approximately independent SNPs. Compared to LD pruning, LD clumping selects the independent SNPs after $p$ value thresholding. Therefore, SNPs showing stronger associations with the disease are preserved, which is preferred in constructing PRS. We note that some literature referred to C+T as P+T, but we treat them as distinct methods in our following discussion.

It is important to note that for both C+T and P+T, only a portion of independent SNPs are utilized in constructing the PRS model, while other SNPs and LD information are ignored. To further improve the prediction accuracy of genetic risks, many PRS methods have been proposed to incorporate genome-wide SNPs and their LD information, such as LDpred[15], LDpred2[16], sBayesR[17], PRS-CS[18] and SDPR[19]. LDpred imposes a point-normal prior for the SNP effect sizes and infers the posterior mean effect sizes using a Markov Chain Monte-Carlo (MCMC) procedure. LDpred2 was further proposed to increase

[1]Department of Biostatistics, Yale School of Public Health, New Haven, CT, USA. [2]Department of Statistics, Columbia University, New York, NY, USA. [3]Department of Psychiatry, Yale School of Medicine, New Haven, CT, USA. ✉e-mail: hongyu.zhao@yale.edu

computational efficiency and provide more stable results than LDpred in dealing with long-range LD regions and traits of sparse genetic architecture. To allow more general effect size distributions, sBayesR performs Bayesian posterior inference based on a mixture prior of point and three normal distributions that represent SNPs with small, medium, and large effects respectively. SDPR performs Bayesian posterior inference based on a Dirichlet process modeling effect sizes with a mixture of 1000 normal distributions. To reduce the computational burden from the combination of different components in millions of SNPs, PRS-CS places a continuous shrinkage prior to the SNP effect sizes in a Bayesian framework. All these LD-based methods have demonstrated their superior performance in some datasets of complex diseases. However, none of them has a dominant performance over other methods.

Among these PRS methods, P+T, C+T, LDpred, and LDpred2 rely on parameters that need to be specified by users beforehand. Although PRS-CS and sBayesR have options to estimate parameters with an additional layer of prior distributions, users can also specify the parameters themselves. For all PRS methods that require tuning parameters, an external individual-level genotype dataset is needed to evaluate different parameter values and choose the best-performing ones. However, as we mentioned before, individual-level genotype data are less accessible than summary statistics. Besides, it is not efficient to leave out a portion of data just for tuning parameters and to estimate SNP effects with the remaining data, leading to information loss and reduced performance for PRS methods. These concerns motivated us to develop a method that can evaluate the performance of a PRS model based on summary statistics used for model training.

For diseases with a binary phenotype, the area under the receiver operating characteristic (ROC) curve (AUC) is the most commonly used criterion in practice for evaluating PRS[5,20,21]. In 2018, Song et al.[22] proposed an estimator of AUC using only summary statistics. This method makes use of an equivalent definition of AUC, i.e. the probability of a PRS from a random case being larger than a PRS from a random control. Based on this definition, AUC can be approximated by a function of the GWAS summary statistics. This method can tune the parameters of a PRS model with summary statistics from another GWAS.

To maximize the power of identifying loci associated with common diseases, some large consortia have conducted meta-analyses of all accessible studies and released summary statistics from these meta-analyses. These summary statistics are usually used as training data to optimize the prediction power of PRS models. In this situation, it is difficult to gain access to summary statistics from another independent GWAS. This problem can not be well addressed if we simply plug the summary statistics from the training data into the derived AUC function, because the variants with larger effects tend to have their effect sizes overestimated and these variants have a larger influence on the PRS than the variants exhibiting small effects. This phenomenon is known as overfitting[23]. If we use the observed effects directly, the overfitting would lead to an inflated predicted value of the AUC and the incorrectly selected values of the parameters.

Built on Song's method, we propose PRStuning, a method that requires only summary statistics from the training data to predict the conventional AUC that needs to be evaluated on another individual-level genotype dataset. We incorporate empirical Bayes (EB) theory to shrink the effect sizes of SNPs, which leads to the attenuation of the predicted AUC so as to overcome the overfitting phenomenon[24]. In PRStuning, we adopt a point-normal mixture model as the prior distribution of SNP effects and estimate the parameters in the model with GWAS summary statistics from the training data. There are two settings depending on the dependency across the selected SNPs used for training the PRS model. When the SNPs are independent, e.g., the SNPs used in P+T, we utilize an expectation-maximization (EM) algorithm to estimate the parameters in the prior distribution and calculate the

posterior distribution of the AUC based on a closed-form formula. When SNPs are dependent due to LD, we use a Gibbs-sampling-based State-Augmentation for Marginal Estimation (SAME) algorithm[25] to estimate the parameters in the model and obtain the Monte-Carlo (MC) samples of the predicted AUC. Once this is accomplished, we can select the parameter values for the PRS method with the best predicted AUC.

We applied PRStuning to GWAS datasets of four common diseases, including coronary artery disease (CAD), type 2 diabetes (T2D), inflammatory bowel disease (IBD), and breast cancer (BC), with four PRS methods, namely P+T, C+T, LDpred, and LDpred2. Results from extensive simulations and real data applications demonstrate that PRStuning can accurately predict the PRS performance across PRS methods and parameters, and it can help with parameter selections.

## Results

### Overview of PRStuning

Define $g_{i,m} \in \{0, 1, 2\}$ as the genotype score of SNP $m$ for individual $i$. A PRS for individual $i$ is the sum of the genotypes $g_i = (g_{i,1}, ..., g_{i,M})$ weighted by the corresponding effects $\omega = (\omega_1, ..., \omega_M)$, i.e.,

$$PRS_i = \sum_{m=1}^{M} \omega_m g_{i,m}. \tag{1}$$

Here $M$ is the total number of pre-selected SNPs used for constructing PRS. Please note that not all SNPs collected in the training GWAS data are necessarily used in PRS calculation. Some PRS methods, such as P+T, select SNPs based on criteria unrelated to association strengths. For those methods, we just need to consider the selected SNPs in estimating AUC. However, some other PRS methods incorporate SNP selection steps based on the associations of the SNPs with the disease, resulting in the inflation of their observed association effects[8,16,17]. For those methods, we consider the SNPs used before the selection step to address the effect size inflation issue with the Empirical-Bayes-based method introduced later. Here we define the pre-selected SNPs as the SNPs used in building the PRS model before running any selection step related to association strengths. For example, the pre-selected SNPs in C+T are actually genome-wide SNPs collected in the training GWAS data and the LD clumping procedure used in C+T is a selection step based on the observed association strength. In this situation, we have $\omega_m = 0$ for SNPs not selected for building PRS in C+T. In contrast, LD pruning is a selection step unrelated to SNP associations with the disease. Therefore, the pre-selected SNPs in P+T are the SNPs selected after an LD pruning step. Different PRS methods have been proposed to estimate the weight vector $\omega$ from a GWAS dataset or its summary statistics for the disease of interest. Here and after we regard $\omega$ as the inferred values from the PRS method of interest.

Based on the definition of AUC and the distribution of PRS, Song[22] formulated AUC as

$$AUC = \Phi(\Delta), \tag{2}$$

where

$$\Delta := \frac{2\sum_{m=1}^{M} \omega_m \delta_m}{\sqrt{\tau_0^2 + \tau_1^2}} \text{ and } \tau_j^2 = \sum_{m=1}^{M} \omega_m^2 s_{j,m}^2 + 2\sum_{m_1 < m_2} \omega_{m_1} \omega_{m_2} R_{m_1,m_2} s_{j,m_1} s_{j,m_2},$$

$$\tag{3}$$

where $j = 0$ indicates controls and 1 indicates cases. Here for SNP $m$, we use $f_{j,m}$ to denote the frequency of the reference allele, $s_{j,m}^2 := 2f_{j,m}(1 - f_{j,m})$ to denote the variance of the genotype, and $\delta_m := f_{1,m} - f_{0,m}$ records the difference between the allele frequencies of the cases and controls, and $\Phi(\cdot)$ is the cumulative distribution

function of a standard normal distribution. We use $R_{m_1,m_2}$ to denote the LD coefficient between SNP $m_1$ and SNP $m_2$.

We can calculate $\tau_j^2$ ($j = 0, 1$) by directly plugging in the observed values of allele frequencies and LD coefficients since $\tau_j^2$ is not directly related to the SNPs' effects on the disease. The observed allele frequencies can be obtained from summary statistics of the GWAS, and LD information can be extracted from another genotype dataset. Some large projects such as the 1000 Genomes project (1KG)[26] and the HapMap3 project (HM3)[27] have made their data publicly available and we can use them as reference panels to calculate the LD coefficients.

For $\delta_m$ in Eq. (3), if we directly plug in the observed allele frequencies $\hat{f}_{0,m}$ and $\hat{f}_{1,m}$ from GWAS, the SNPs exhibiting large allele frequency differences tend to have their effect sizes overestimated, and these SNPs have larger contributions to the PRS than the SNPs showing smaller effects. The overfitting of the SNP effects would lead to an inflated predicted value of the AUC and incorrectly selected values of the parameters. Therefore, we adopt an Empirical Bayes method in PRStuning to shrink the effects so as to reduce the influence of overfitting. In the Supplementary Methods section, we provide a theoretical demonstration of how overfitting happens and the rationale of alleviating overfitting with a Bayes estimator.

In GWAS, $z$-scores from the allele frequency difference test are usually used to assess the association of each SNP with the disease. Each $z$-score is calculated with the following formula:

$$z_m = \frac{\hat{f}_{1,m} - \hat{f}_{0,m}}{\sqrt{s_{1,m}^2/4n_1 + s_{0,m}^2/4n_0}}, \qquad (4)$$

where $\hat{f}_{j,m}$ is the observed allele frequency for each group, $s_{j,m}^2$ is the variance of the genotype in the controls or cases, and $n_0, n_1$ are the sample sizes of the two groups. To simplify this expression, we define $s_m := \sqrt{s_{1,m}^2/4n_1 + s_{0,m}^2/4n_0}$. Based on this definition, we have $z_m|\delta_m \sim N(\delta_m/s_m, 1)$ given the allele frequency difference $\delta_m$.

Here we denote the allele frequencies among controls and cases when SNP $m$ is assumed to be independent with other SNPs as $p_{0,m}$ and $p_{1,m}$, respectively. Note that $f_{j,m}$ is the allele frequency of SNP $m$ marginalizing over other SNPs, which is different from $p_{j,m}$ ($j = 0, 1$). We use $\beta_m$ to denote the underlying effect of SNP $m$ in terms of changing allele frequencies between controls and cases, i.e., $\beta_m = p_{1,m} - p_{0,m}$. If SNP $m$ has no risk on the disease, we have $\beta_m = 0$. Let $\beta = (\beta_1, ..., \beta_M)$. In the Supplementary Methods section, we further demonstrate that the marginalized allele frequency difference $\delta = (\delta_1, ..., \delta_M)$ is related to the LD pattern among the pre-selected SNPs and $\beta$, i.e.,

$$\delta = SRS^{-1}\beta, \qquad (5)$$

where $S$ is a diagonal matrix with the $m$-th diagonal element equal to $s_m$, and $R$ is the LD coefficient matrix. Given $\delta$, the joint distribution of the $z$-scores $z = (z_1, ..., z_M)$ is

$$z|\delta \sim N(S^{-1}\delta, R). \qquad (6)$$

We further assume that the standardized effect $\beta_m/s_m$ follows a point-normal distribution, i.e.,

$$\frac{\beta_m}{s_m} \overset{iid}{\sim} (1 - \pi)\delta_0 + \pi N(0, \sigma^2). \qquad (7)$$

Here $\delta_0$ is a point mass at zero, $\pi$ represents the prior proportion of SNPs that have an effect on the disease, and $\sigma^2$ is the variance of $\beta_m/s_m$ in the risk SNPs. This point-normal distribution is also used in LDpred as the prior distribution. The relationship between $\sigma^2$ and the heritability of the disease is presented in Section "Notations and

assumptions" and the Supplementary Methods section. With this assumption, we derived an expectation-maximization (EM) algorithm to estimate $(\pi, \sigma^2)$ and calculated the posterior distribution of the AUC when pre-selected SNPs are independent. When SNPs are linked by LD, we derived a Gibbs-sampling-based SAME algorithm to estimate $(\pi, \sigma^2)$ and obtained the MC samples of the predicted AUC. Once this is accomplished, we can select the parameter values for the PRS method with the best predicted AUC. Details of PRStuning are presented in Section "Methods".

## Simulation experiments

For our simulation experiments, we considered predicting the performance and tuning the parameters for four commonly used PRS methods, namely, P+T, C+T, LDpred, and LDpred2. In the experiments, we varied the $p$-value thresholds for P+T and C+T from {1, 5e − 1, 5e − 2, 5e − 3, 5e − 4, 5e − 5, 5e − 6}. In P+T, $p$-value threshold=1 means that no further filtering step based on $p$-values is utilized on pre-selected approximately independent SNPs after LD pruning. In C+T, $p$-value threshold=1 means we conduct LD clumping on genome-wide SNPs without filtering based on $p$-values. While for LDpred, we chose the proportion of the risk SNPs $\pi$ from {1, 3e − 1, 1e − 1, 3e − 2, 1e − 2, 3e − 3, 1e − 3, 3e − 4, 1e − 4, 3e − 5, 1e − 5}. This is the default setting of LDpred. Because LDpred2 had convergence issues when the risk SNP proportion was set to an extremely small value for simulations based on simulated genotype data, we varied $\pi$ from {1, 6e − 1, 3e − 1, 1e − 1, 6e − 2, 3e − 2, 1e − 2} which had a smaller range but finer resolution than the set used in LDpred. For simulations based on real genotype data, we used the same parameters as LDpred.

There are two purposes of our method: to predict the AUC and to select tuning parameters. In our experiments, we used another independent dataset with individual-level genotype data as testing data. The AUC of the PRS assessed on the testing data and the parameters showing the best prediction performances on the testing data were treated as benchmarks. To evaluate the performance of PRStuning, we evaluated the performance of PRStuning with two measures: the correlation of the AUC estimates ($\rho_{AUC}$) and the relative difference of the highest AUC estimates ($rd_{AUC}$). We define $\rho_{AUC}$ as the correlation of the PRStuning-predicted AUC values and those estimated on the testing data. A high value of $\rho_{AUC}$ indicates that the predicted AUC using our method is highly correlated with the AUC on the testing data. We define $rd_{AUC}$ as the relative difference between the predicted AUC with the best-performing parameter tuned by PRStuning and the AUC with the best-performing parameters on the testing data. Here best-performing parameters are defined as those achieving the highest AUC values. A small value of $rd_{AUC}$ indicates that the tuning parameter selected by PRStuning and the actual best-performing parameter have comparable performances. These two metrics are complementary to each other in the sense that, $\rho_{AUC}$ measures how much the AUC patterns across parameter values for PRStuning and testing data align with each other, while $rd_{AUC}$ measures the point difference between the highest AUC values for the two methods. Therefore, we would like to evaluate the results with both metrics.

We first consider the case where the pre-selected SNPs are independent. In our simulations, we set the prevalence of the disease to $\kappa = 1\%$. For each SNP, we simulated its allele frequency in the general population based on a uniform distribution $U(0.05, 0.95)$. Then we generated its risk effects on the disease based on the two-component mixture model Eq. (7), in which we set the proportion of the risk SNPs to $\pi = 0.05$ and the variance of the risk effects to $\sigma^2 = 0.001n$. Here $n$ is the total sample size of the GWAS used in the training data. We assume the GWAS is balanced with an equal number of cases and controls. According to the central limit theorem, we have $s_m \propto 1/\sqrt{n}$. Hence it is reasonable to assume $\sigma^2 \propto n$.

In total, we simulated $M = 10,000$ independent SNPs and varied the sample size from 4,000 to 10,000 in the training GWAS to explore

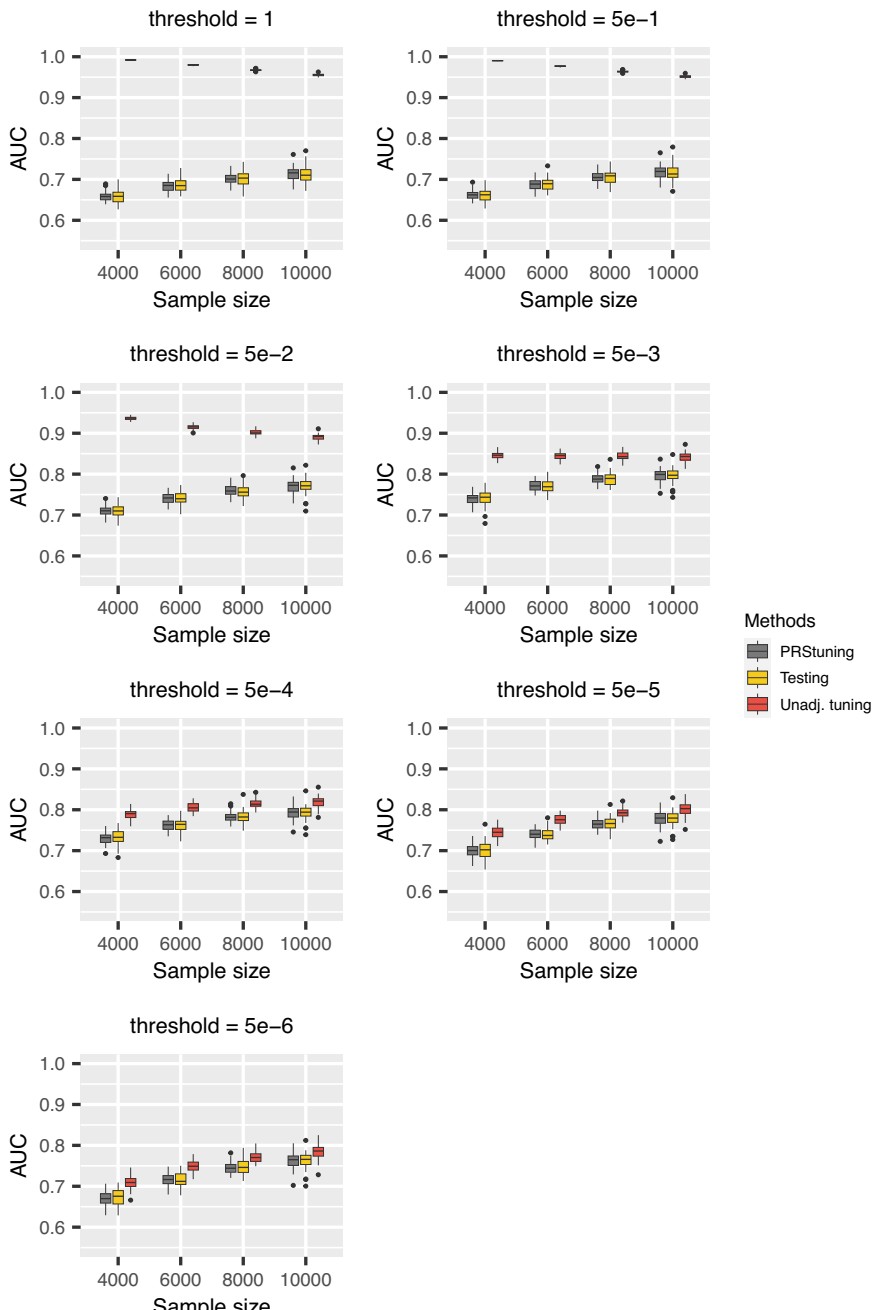

**Fig. 1 | AUC boxplots for P+T in the simulation experiments with independent SNPs.** Each box represents 50 replications and is presented as median values and the first and third quartiles. The upper/lower whisker extends from the hinge to the largest/smallest value at most 1.5 IQR from the hinge. We changed the $p$-value threshold from $\{1, 5e-1, 5e-2, 5e-3, 5e-4, 5e-5, 5e-6\}$ and the sample sizes of training data from 4000 to 10,000. The grey, yellow, and red panels represent AUC predicted from PRStuning, AUC evaluated on testing data, and the unadjusted AUC directly estimated by plugging in the training summary statistics, respectively. The AUC evaluated on the testing data is the benchmark. PRStuning is able to yield AUC estimates comparable to the benchmark results. Source data are provided as a Source Data file.

the performance trend across different sample sizes. Each sample size setting was replicated 50 times. And for each replication, we simulated additional 1000 cases and 1000 controls as testing data. We used the AUC evaluated on the testing data as the benchmark, and compared the AUC predicted by PRStuning and the unadjusted AUC obtained by directly plugging in the training summary statistics with the benchmark. Since all SNPs are independent, we only considered P+T as the PRS method.

Figure 1 shows the boxplots of AUC values corresponding to different $p$-value thresholds and sample sizes of training data for P+T. The grey, yellow, and red panels represent AUC predicted from PRStuning,

AUC calculated from testing data, and the unadjusted AUC obtained by directly plugging in the training summary statistics, respectively. As expected, the unadjusted AUC estimates were inflated compared to the benchmark due to the overfitting problem. In contrast, with the same summary statistics from the training data, PRStuning was able to shrink the estimates of allele frequency differences and produce AUC estimates comparable to those from the testing data.

In order to further demonstrate the accuracy of PRStuning, we summarize the average correlation of the AUC estimates $\rho_{\mathrm{AUC}}$ and the average relative difference of the best-performing AUC estimates $rd_{\mathrm{AUC}}$ in Table 1. Those metrics are complementary to each other since

**Table 1 | Summary of the average values of $\rho_{AUC}$ and $rd_{AUC}$ in the simulation experiments with independent SNPs**

| Metric | $n = 4000$ | $n = 6000$ | $n = 8000$ | $n = 10{,}000$ |
|---|---|---|---|---|
| $\rho_{AUC}$ | 0.976 | 0.988 | 0.993 | 0.996 |
| $rd_{AUC}$ | 1.3% | 1.0% | 0.9% | 0.7% |

We considered P+T as the PRS method. For each sample size, 50 replications were generated in the experiment.

two vectors can be perfectly correlated but differ a lot. The values of $\rho_{AUC}$ were at least 0.976, which indicates that PRStuning is capable of accurately predicting the AUC pattern on testing data. Moreover, the average values of $rd_{AUC}$ were at most 1.3%, indicating that PRStuning can effectively select parameter values that achieve performance comparable to the best-performing parameter in the testing data. Note that $\rho_{AUC}$ increased and $rd_{AUC}$ decreased as the sample size of training GWAS increased. This is expected because a larger sample size in the training data can lead to higher accuracy in estimating allele frequency differences.

We also evaluated PRStuning when the training and testing data are heterogeneous. To be more specific, we considered two different scenarios. In the first scenario, we assumed that the allele frequencies from the training and testing data were different and the difference between allele frequencies was generated from $N(0, 0.01^2)$. In the other scenario, we assumed that the effect sizes between training and testing data were different and the difference between effects of risk SNPs followed $N(0, 0.0005n)$. The results of these experiments for P+T are provided in the Supplementary Figures 3-4. The figures demonstrate that PRStuning can still estimate the AUC well when the pooled allele frequencies are different between the training and testing data. However, if the effects of risk SNPs are different between training and testing data, the AUC from PRStuning was overestimated, leading to inaccurate performance to tune parameters.

We then considered the case where the pre-selected SNPs are not filtered by any independence criterion for SNPs. In this case, the pre-selected SNPs are linked as reflected in their LD. We first performed simulations with SNPs with an AR(1) auto-regressive LD structure. We fixed the auto-regressive coefficient $\rho$ to 0.2, which is the correlation coefficient between two adjacent SNPs. Similar to the simulation scenario with independent SNPs, we simulated the reference allele frequencies in the population from $U(0.05, 0.95)$, and the risk effects from a point normal distribution Eq. (7), in which $\pi = 0.05$ and $\sigma^2 = 0.0005n$. The variance of risk effects is proportional to the sample size of the GWAS since $s_m \propto 1/\sqrt{n}$ according to the central limit theorem.

We varied the sample size from $4,000$ to $10,000$ in the training GWAS and generated 50 replications for each sample size. We used CorBin[28], an R package for generating high dimensional binary data with a specific correlation structure, to generate individual-level genotype data. Specifically, we generated 1,000 cases and 1,000 controls as testing data for each replication. We additionally simulated 1,000 samples as a reference panel for calculating LD coefficients. We used both C+T and LDpred as the PRS methods in this experiment. In LDpred, we need to specify another parameter named LD radius, which is the number of SNPs on each side of a given SNP for computing pairwise LD. The LD radius was set to 5, indicating that the SNPs used for computing LD have pairwise correlations above $0.2^5 \approx 3 \times 10^{-4}$ based on the AR(1) LD structure.

To demonstrate the predictive accuracy of PRStuning, we again regarded the AUC evaluated on the testing data as the benchmark and compared the AUC predicted by PRStuning and the unadjusted AUC with the benchmark. Figures 2, 3 and Supplementary 1 demonstrate the AUC boxplots for C+T, LDpred, and LDpred2 with different parameter values, respectively. For both PRS methods, the unadjusted AUC

estimates were largely overestimated compared to the benchmark due to overfitting. On the contrary, the AUC estimates predicted by PRStuning were very close to the benchmark results, especially when the sample size became large.

We summarize the average values of $\rho_{AUC}$ and $rd_{AUC}$ for C+T and LDpred in Table 2. For both C+T and LDpred, the average values of $\rho_{AUC}$ were at least 0.754 in all sample size settings, indicating PRStuning can accurately predict the AUC on testing data. The average values of $rd_{AUC}$ were below 3.1%, meaning PRStuning can effectively select a parameter that achieves performance comparable to the actual best-performing parameter on the testing data. Again, we can observe an increasing tendency in $\rho_{AUC}$ and a decreasing tendency in $rd_{AUC}$ as we increase the sample size of the training GWAS as the result of the increase in estimation accuracy of the allele frequency differences.

We evaluated PRStuning when the training and testing data were heterogeneous, where we considered three different scenarios. In the first scenario, we assumed the allele frequencies from training and testing data were different and the differences between allele frequencies were generated from $N(0, 0.01^2)$. In the second scenario, we assumed that the effect sizes between training and testing data were different and the difference between effects of risk SNPs followed $N(0, 0.0002n)$. In the third scenario, the LD structure of the testing data was AR(1) with an auto-regressive coefficient $\rho = 0.15$, which is different from the auto-regressive coefficient of the training data. The results of these experiments for C+T, LDpred, and LDpred2 are provided in Supplementary Figures 5-13. Generally speaking, the figures demonstrate that PRStuning can still estimate the AUC well when the pooled allele frequency and LD matrix are different between training and testing data. However, if the effects of risk SNPs are different between training and testing data, the AUC from PRStuning was overestimated, leading to inaccurate performance to tune parameters.

To investigate whether including more individuals in the reference panel can improve the performance of PRStuning, we conducted simulation experiments to compare its performance with the performance based on the ground truth LD matrix. The comparison results using C+T, LDpred, and LDpred2 to construct PRS can be found in Supplementary Figures 14-16, respectively. From the figures, we observe that the performance of PRStuning based on the LD matrix estimated from 1,000 individuals was almost the same as the performance based on the ground truth LD matrix. Thus, with a sufficient number of individuals in the reference panel, there may be little improvement in performance by including more individuals in the LD matrix calculation.

To further demonstrate the effectiveness of PRStuning, we calculate the sensitivity of the PRS model tuned by PRStuning, which is the proportion of true cases among predicted ones from the PRS model. The cutoff value for PRS is selected by Youden's J statistic, which is defined as the sum of sensitivity and specificity minus one and is the most commonly used criterion to select the cutoff value for a binary classifier[29]. The true case proportions of simulation experiments for the four PRS methods are summarized in Supplementary Figure 2.

We also evaluated PRStuning with simulations based on real genotype data. The experiments were conducted based on genotype data collected from the UK Biobank (UKBB)[30], which collected genetic and health records from around $500,000$ participants in the UK. The quality control procedure is summarized in the Supplementary Methods section. We only selected independent individuals with European ancestry in the experiments. Since only SNPs presented in the HapMap 3 project (HM3 SNPs) were used in the reference panel for reliable LD estimation and computation efficiency, we focused on the SNPs in HM3 in the UKBB dataset. This resulted in a total of $1,027,699$ HM3 SNPs and $272,751$ individuals passing the quality control criteria.

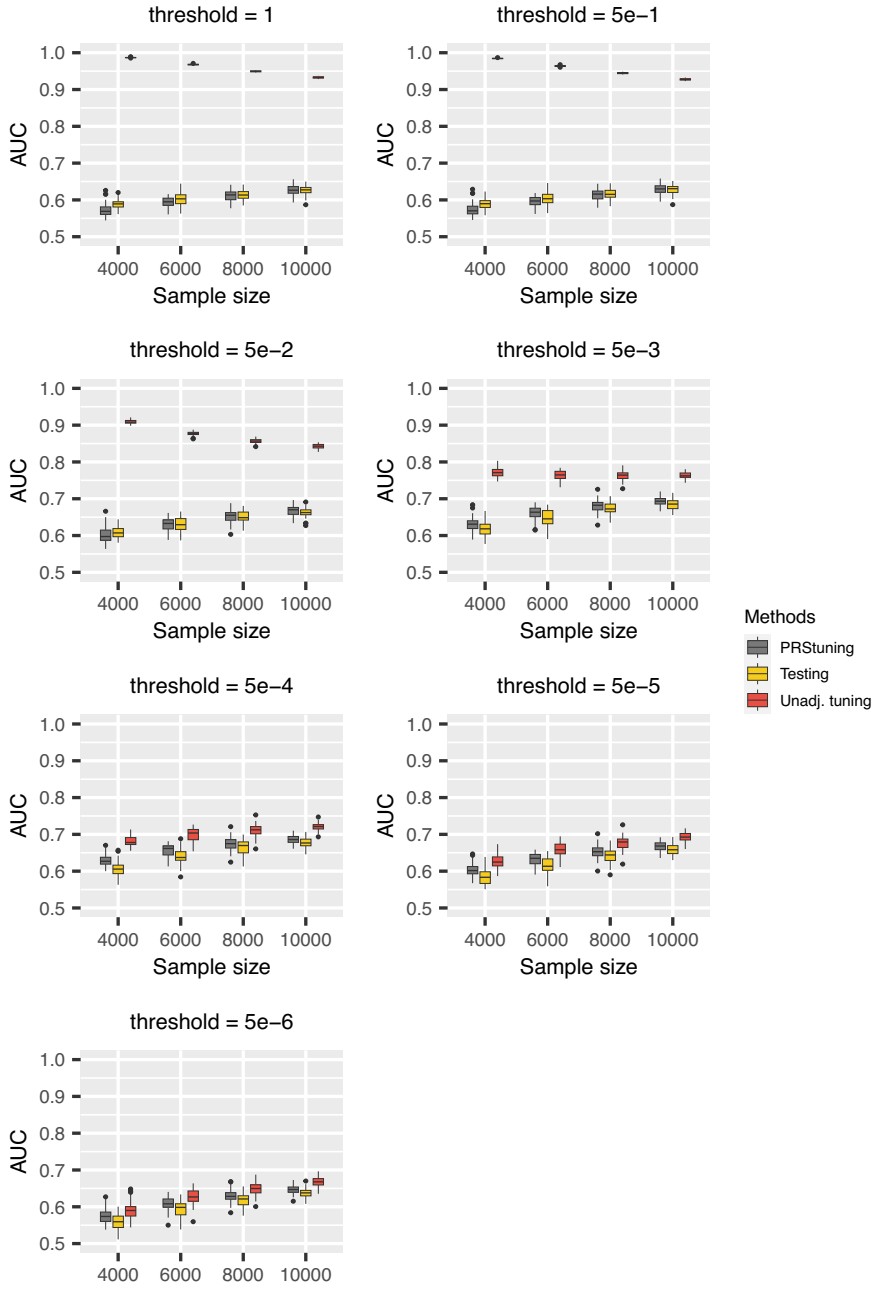

**Fig. 2 | AUC boxplots for C+T in the simulation experiments with correlated SNPs.** Each box represents 50 replications and is presented as median values and the first and third quartiles. The upper/lower whisker extends from the hinge to the largest/smallest value at most 1.5 IQR from the hinge. We changed the *p*-value threshold from {1, 5e − 1, 5e − 2, 5e − 3, 5e − 4, 5e − 5, 5e − 6} and the sample sizes of training data from 4000 to 10,000. The grey, yellow, and red panels represent AUC predicted from PRStuning, AUC evaluated on testing data, and the unadjusted AUC directly estimated by plugging in the training summary statistics, respectively. The AUC evaluated on the testing data is the benchmark. PRStuning is able to yield AUC estimates comparable to the benchmark results. Source data are provided as a Source Data file.

We used the two-component mixture model Eq. (7) to simulate risk effects for SNPs with $\pi = 0.1\%$ and $\sigma^2 = 0.04$. The phenotypes of the individuals were simulated based on the additive assumption. Among all individuals, we randomly selected 80% of them for GWAS analysis to calculate the summary statistics as training data and the rest as testing data. We used the data collected from the 1000 Genomes Project (1KG)[26] as the reference panel for calculating LD. In the experiments, we used both C+T and LDpred as the PRS methods and compared the AUC estimates predicted by PRStuning with the values calculated on the testing data. The LD radius to be specified in LDpred was set to $M/3000 \approx 343$, which is the default practice

suggested by LDpred and corresponds to a 2Mb LD window on average in the human genome[15].

In Table 3, we summarize the AUC results of C+T, LDpred, and LDpred2 with different parameter values for both PRStuning and testing genotype data. The AUC estimates from PRStuning were very close to the actual AUC values obtained from the testing data. For C+T, the correlation $\rho_{AUC}$ reached 0.994, the relative difference $rd_{AUC}$ was 3.8%, and the sensitivity of the tuned PRS model based on PRStuning was 80.6%. For LDpred, $\rho_{AUC}$ reached 0.998, $rd_{AUC}$ was just 1.3%, and the sensitivity was 74.8%. It is worth noting that PRStuning was able to detect the dramatic decrease in the testing performance of LDpred

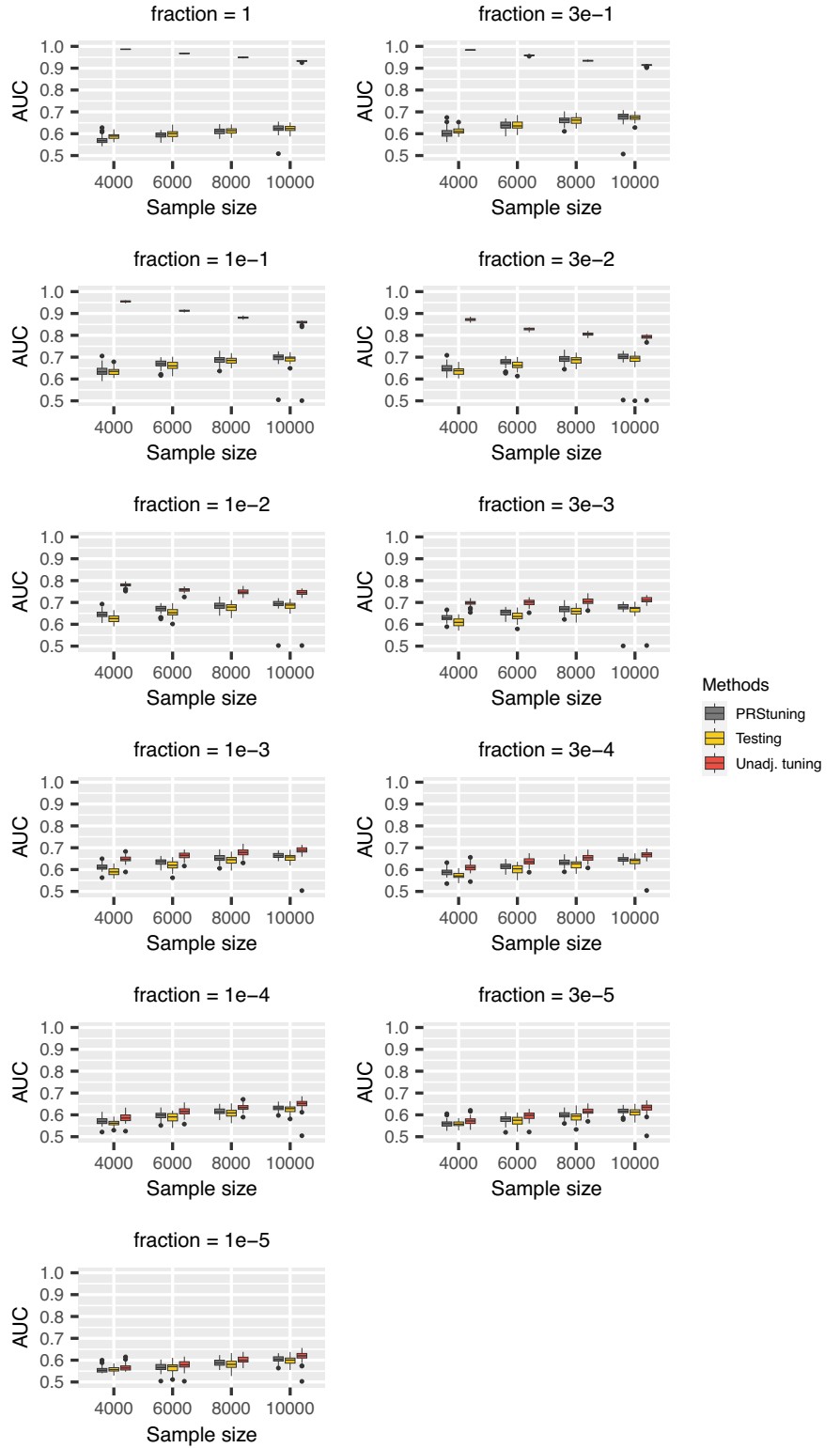

**Fig. 3 | AUC boxplots for LDpred in simulation experiments with correlated SNPs.** Each box represents 50 replications and is presented as median values and the first and third quartiles. The upper/lower whisker extends from the hinge to the largest/smallest value at most 1.5 IQR from the hinge. We changed the proportion of risk SNPs from {1, 3e − 1, 1e − 1, 3e − 2, 1e − 2, 3e − 3, 1e − 3, 3e − 4, 1e − 4, 3e − 5, 1e − 5} and the sample sizes of training data from 4000 to 10,000. The grey, yellow, and red panels represent AUC predicted from PRStuning, AUC calculated from testing data, and the unadjusted AUC, respectively. Source data are provided as a Source Data file.

when $\pi$ was dropped from 1e − 1 to 3e − 2. For LDpred2, $\rho_{AUC}$ reached 0.989, $rd_{AUC}$ was 7.0%, and the sensitivity was 85.3%. These results further suggest the accuracy in AUC estimation and effectiveness in parameter tuning using PRStuning on SNPs linked by LD.

## Real data applications

We applied PRStuning to GWAS summary statistics from four diseases, including coronary artery disease (CAD), type 2 diabetes (T2D), and inflammatory bowel disease (IBD). Table 4 summarizes the sources of

the publicly available GWAS summary statistics and their corresponding sample sizes. Note that the summary statistics from all three datasets are results of meta-analyses and the reported sample sizes represent the total numbers of individuals among all aggregated studies. The actual sample size used to calculate the summary statistics of each SNP was less than the reported sample size, since some of the studies may not have genotypes on this SNP.

We used these summary statistics to train the PRS models based on P+T, C+T, LDpred, and LDpred2. Then we used the data collected from the UKBB as the testing data for evaluating the actual prediction performance of the built PRS models. Only the SNPs with minor allele frequencies greater than 5% were included in building the PRS models. Details of the quality control procedure and phenotype extraction method for the UKBB data are provided in the Supplementary Methods section. In line with the simulation experiments based on UKBB genotype data, we only incorporated independent European-ancestry individuals and HM3 SNPs in the UKBB dataset, resulting in 272,751 individuals and 1,027,699 HM3 SNPs. Regardless of which PRS method is considered, only the SNPs overlapped between GWAS summary statistics and the testing data were considered in our analyses. The numbers of the overlapping SNPs for these diseases are summarized in Table 4.

In PRStuning, we adopted the EM algorithm 4.2 for PRS models built by P+T since the pre-selected SNPs were approximately independent, and the Gibbs sampling-based SAME algorithm 4.3 for C+T and LDpred due to the presence of LD among the pre-selected SNPs. The LD radius in LDpred was set to $M/3000$, which is the default practice suggested by LDpred. Figure 4 shows the predicted AUC by PRStuning and the actual AUC on testing data for four diseases with different PRS models. The dotted and solid horizontal lines respectively refer to the highest AUC for PRStuning and testing data. It is evident in the figure that the AUC predicted by PRStuning and the AUC calculated from testing data had similar patterns across different parameter values, particularly for LDpred. For CAD, the AUC of LDpred increased when the risk SNP proportion $\pi$ was reduced from 1 to $1e-2$. It peaked at $1e-2$ and then started to decrease when we kept reducing the value of $\pi$. This pattern was exactly predicted by PRStuning. More complex patterns of AUC were observed for LDpred in T2D and CAD. The AUC values in both diseases had double modes across parameter values. For T2D, the AUC of LDpred peaked at $3e-2$ and $3e-4$. For IBD, the AUC of LDpred peaked at $3e-2$ and $1e-5$. Still, PRStuning predicted the exact same patterns of AUC for both diseases. This demonstrates the high predictive accuracy of PRStuning. More detailed information for the predicted AUC by PRStuning and the actual AUC on testing data is summarized in Supplementary Table 2.

To further explain why there were double modes for AUC with different parameter values, we refer back to the calculation of Δ in Eq. (3) since AUC is monotonically increasing with respect to Δ. The numerator of Δ is a linear combination of the weights $\omega = (\omega_1, \ldots, \omega_M)^T$ used in PRS, whereas the denominator is the square root of a quadratic function of $\omega$, which can be further expressed as

$$\sqrt{\tau_0^2 + \tau_1^2} = \sqrt{\omega^T(S_0 R S_0 + S_1 R S_1)\omega}, \tag{8}$$

### Table 2 | Summary of the average values of $\rho_{AUC}$ and $rd_{AUC}$ in the simulation experiments with correlated SNPs for C+T and LDpred

| PRS | Metric | $n = 4000$ | $n = 6000$ | $n = 8000$ | $n = 10,000$ |
|---|---|---|---|---|---|
| C+T | $\rho_{AUC}$ | 0.622 | 0.832 | 0.904 | 0.951 |
| | $rd_{AUC}$ | 2.7% | 2.5% | 1.6% | 1.6% |
| LDpred | $\rho_{AUC}$ | 0.876 | 0.955 | 0.970 | 0.977 |
| | $rd_{AUC}$ | 2.9% | 2.3% | 1.7% | 1.6% |

For each sample size, 50 replications were generated in the experiment.

### Table 3 | The predicted AUC values for C+T, LDpred, and LDpred2 with different parameters in the simulation experiment based on the UKBB data

| C+T | | | | | | | | | | | |
|---|---|---|---|---|---|---|---|---|---|---|---|
| Threshold | 1 | $5e-1$ | $5e-2$ | $5e-3$ | $5e-4$ | $5e-5$ | $5e-6$ | $5e-7$ | $5e-8$ | | |
| PRStuning | 0.789 | 0.790 | 0.816 | 0.830 | 0.834 | 0.835 | 0.835 | 0.835 | 0.835 | | |
| Testing | 0.793 | 0.795 | 0.830 | 0.852 | 0.860 | 0.865 | 0.867 | 0.868 | 0.868 | | |
| **LDpred** | | | | | | | | | | | |
| $\pi$ | 1 | $3e-1$ | $1e-1$ | $3e-2$ | $1e-2$ | $3e-3$ | $1e-3$ | $3e-4$ | $1e-4$ | $3e-5$ | $1e-5$ |
| PRStuning | 0.747 | 0.784 | 0.732 | 0.593 | 0.550 | 0.532 | 0.519 | 0.513 | 0.509 | 0.504 | 0.502 |
| Testing | 0.790 | 0.813 | 0.749 | 0.588 | 0.548 | 0.527 | 0.518 | 0.509 | 0.506 | 0.500 | 0.506 |
| **LDpred2** | | | | | | | | | | | |
| $\pi$ | 1 | $3e-1$ | $1e-1$ | $3e-2$ | $1e-2$ | $3e-3$ | $1e-3$ | $3e-4$ | $1e-4$ | $3e-5$ | $1e-5$ |
| PRStuning | 0.768 | 0.777 | 0.767 | 0.781 | 0.767 | 0.837 | 0.890 | 0.919 | 0.875 | 0.915 | 0.926 |
| Testing | 0.780 | 0.787 | 0.793 | 0.785 | 0.771 | 0.818 | 0.846 | 0.859 | 0.838 | 0.856 | 0.861 |

We randomly selected 80% of individuals as training data and the rest as the testing data. The data from the 1KG were used as the reference panel. For C+T, LDpred, and LDpred2, respectively, the correlation values $\rho_{AUC}$ reached 0.994, 0.998, 0.989, the relative difference values $rd_{AUC}$ were 3.8%, 1.3%, 7.0%, and the sensitivity values of the tuned PRS model based on PRStuning were 80.6%, 74.8%, 85.3%. Sensitivity is calculated using the cutoff value selected by Youden's J statistic.

### Table 4 | Summary of the publicly available GWAS summary statistics used in real data applications

| Disease | Source | Sample Size | #SNPs | #Overlapping SNPs (UKBB, 1KG, HM3) |
|---|---|---|---|---|
| Type 2 Diabetes (T2D) | DIAGRAM[51] | $n_O = 56,962$ $n_1 = 12,171$ | 1,938,21 | 718,340 |
| Coronary Artery Disease (CAD) | CARDIoGRAM[52] | $n_O = 64,762$ $n_1 = 22,233$ | 2,121,277 | 861,825 |
| Inflammatory Bowel Disease (IBD) | IIBDGC[53] | $n_O = 38,155$ $n_1 = 48,485$ | 4,911,413 | 952,376 |
| Breast Cancer (BC) | BCAC[54] | $n_O = 17,588$ $n_1 = 14,910$ | 11,050,495 | 1,016,333 |

The sources of GWAS summary statistics, their sample sizes, and the SNP numbers are presented in the table. We also report the number of overlapping SNPs among UKBB, 1KG, and HM3. These SNPs were used in PRStuning.

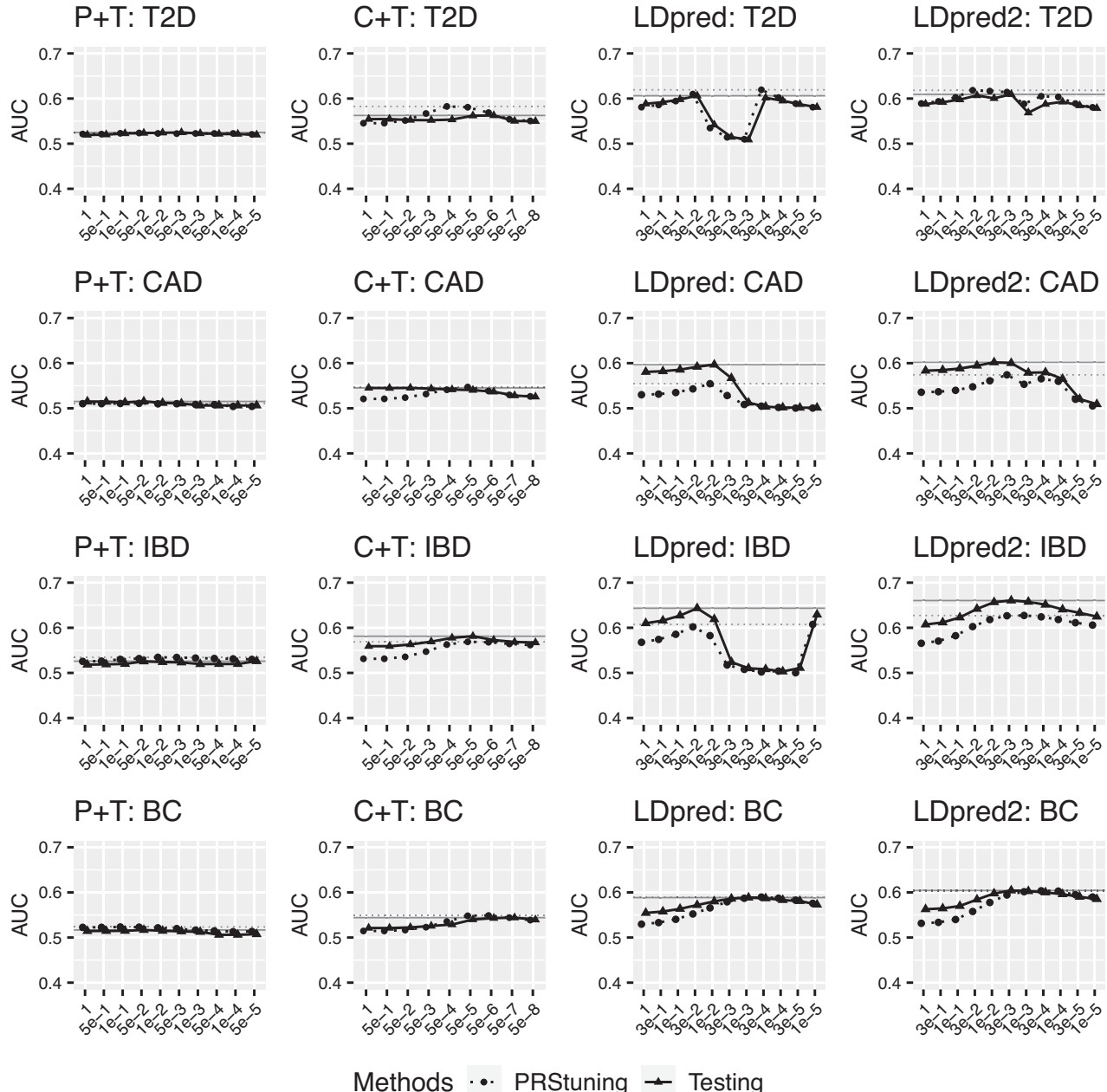

**Fig. 4 | The predicted AUC by PRStuning and the actual AUC on testing data for four diseases with PRS models built from P+T, C+T, LDpred, and LDpred2 using different parameters.** The four panels present the results of P+T, C+T, LDpred, and LDpred2, respectively. The dotted and solid horizontal lines refer to the highest AUC for PRStuning and testing data. The overall patterns of AUC predicted by PRStuning and calculated from testing data across different parameter values were similar. Detailed AUC values for different methods and tuning parameters are summarized in Supplementary Table 2. Source data are provided as a Source Data file.

where $S_0$ and $S_1$ are diagonal matrices with diagonal elements encoding $(s_{0,1}, …, s_{0,M})$ and $(s_{1,1}, …, s_{1,M})$, respectively. The weights in the PRS model were calculated based on different values of parameters. In Supplementary Figure 17, we demonstrate the denominators and numerators of $\Delta$ with different parameter values in LDpred for the four diseases. From the figure, we can observe that both the denominator and numerator were actually unimodal functions with respect to the parameter values that peak at different parameter values. Their ratio led the $\Delta$ to become bimodal functions with respect to the parameter values.

In Figure 4, we do observe some underestimation of AUC for C+T, LDpred, and LDpred2 on CAD and IBD. This is because the summary statistics collected are results of meta-analyses. The actual sample size used for calculating the summary statistics of each SNP is less than the reported sample size, because some of the studies may not have genotypes at this SNP. Some consortia, such as GLGC[31], provide the sample size used for calculating summary statistics of each SNP, but most consortia do not provide this information. Even if we have the sample size for each SNP, we can not infer the number of non-overlapping individuals for calculating summary statistics of two SNPs. The non-overlapping individuals will change the correlations between $z$-values. In our analysis, we simply plugged the total sample sizes reported by the summary statistics into PRStuning. According to Eq. (16), the inflation of the sample size would lead to the systematic underestimation of $s_m$. Based on Eq. (2), we know that AUC is monotonically increasing with respect to $\Delta$, and we have $\Delta \propto$

**Table 5 | Summary of $\rho_{AUC}$ and $rd_{AUC}$ when using PRStuning to predict AUCs for four PRS methods on four diseases**

| Disease | P+T | C+T | LDpred | LDpred2 |
|---|---|---|---|---|
| T2D | 0.731 (0.2%) | 0.514 (3.5%) | 0.982 (2.2%) | 0.856 (1.5%) |
| CAD | 0.817 (0.8%) | −0.102 (0.4%) | 0.969 (7.1%) | 0.784 (4.6%) |
| IBD | 0.491 (1.6%) | 0.858 (2.1%) | 0.987 (5.6%) | 0.926 (5.0%) |
| BC | 0.936 (1.2%) | 0.956 (0.9%) | 0.950 (0.2%) | 0.922 (0.1%) |

The $rd_{AUC}$ values are summarized in parenthesis. Note that the standard deviations among the AUC values with different parameters were less than 0.01 for both methods when using C+T on CAD. The extremely small standard deviations of AUC contribute to the large variation of the correlation, leading to a negative $\rho_{AUC}$.

$\sum_{m=1}^{M} \omega_m \delta_m$ and $\delta = SRS^{-1}\beta$. We estimate $S^{-1}\beta$ directly from $z$-scores, which are not influenced by the underestimation of $s_m$. Therefore, the underestimation of $s_m$ would further lead to the underestimation of AUC.

To further illustrate the predictive accuracy of PRStuning, we calculated $\rho_{AUC}$ and $rd_{AUC}$ with different PRS methods for the four diseases. The results of $\rho_{AUC}$ and $rd_{AUC}$ are summarized in Table 5. The low values of $rd_{AUC}$ indicate that the prediction performance under the PRStuning-selected parameter approximated the best performance on the testing data accurately, especially for C+T and P+T. Even though LDpred had higher $rd_{AUC}$ compared to the other two PRS methods, it yielded values of $\rho_{AUC}$ all above 0.95. The high values of $\rho_{AUC}$ indicate that PRStuning can accurately predict the pattern of AUC with respect to the parameters on the testing data. This can be clearly observed from Figure 4. These results show that PRStuning can help us select the best-performing parameters in PRS methods with only summary statistics from the training data.

We note that the correlation between AUC predicted by PRStuning and calculated from the testing data was negative with C+T for CAD. However, also note that the standard deviations among the AUC values with different parameters for both methods were less than 0.01 in this scenario. The extremely small standard deviations of AUC contribute to the large variation of the correlation. Therefore, the correlation is relatively uninformative in characterizing the relationship between the predicted and the actual AUC values. On the other hand, the small value of $rd_{AUC}$ (0.4%) suggests the effectiveness of PRStuning. The sensitivity values of the tuned PRS model based on PRStuning and Youden's J statistic are summarized in Supplementary Table 3.

We also compared PRStuning with PUMAS[32], a method to estimate predictive $R^2$ for PRS models using summary statistics from GWAS by sampling pseudo-summary-statistics. To compare predictive $R^2$ with AUC, we first converted Pearson's correlation to Spearman's rank correlation and then linearly mapped the latter to AUC[33]. In Supplementary Table 4, we summarize $\rho_{AUC}$ and $rd_{AUC}$ based on PUMAS. We observe that PRStuning outperformed PUMAS across all real data and PRS methods, and that PUMAS is especially incapable of predicting the AUC well for LDpred and LDpred2.

## Discussion

PRS methods have been proven useful for the prediction of common disease risks, which can help improve disease prevention and early treatment. Some PRS methods require users to specify the values for parameters. However, to tune the parameters, an external individual-level genotype dataset is often needed to evaluate the prediction performance of different parameter values. However, individual-level genotype data are much less accessible compared to GWAS summary statistics due to privacy and security concerns. Additionally, leaving out partial data for parameter tuning can also reduce the predictive accuracy of the PRS model.

These concerns motivated us to propose PRStuning, an empirical Bayes method that only requires summary statistics from the training GWAS to evaluate PRS and tune the parameters. PRStuning is based on an AUC estimator proposed in[22], which is a function of the GWAS summary statistics. However, plugging the training summary data directly into this estimator would cause overfitting, leading to an inflation of the predicted AUC. To tackle this problem, we adopted the empirical Bayes approach to shrinking the predicted AUC based on the estimated genetic architecture. Extensive simulation experiments and real data applications on four diseases with four PRS methods demonstrated that PRStuning is capable of accurately predicting the AUC on the testing data and selecting the best-performing parameters.

The core of PRStuning is to estimate the allele frequency differences among SNPs. To do so, we need to input the sample sizes of the cases and controls in the training data. Usually, they are provided in the sources of GWAS summary statistics. However, if the summary statistics were derived from a meta-analysis, not all SNPs were genotyped in all studies included in the meta-analysis. In this case, the actual sample sizes used for calculating the summary statistics are less than the reported total sample sizes in the meta-analysis for some SNPs. This may lead to underestimation in AUC according to Eq. (2). This phenomenon was observed when we applied PRStuning to C+T and LDpred on CAD and IBD, where the AUC estimates from PRStuning were lower than the actual values in the testing data. Nevertheless, according to our experimental results, the underestimation phenomenon will not influence the performance of parameter selection since the overall pattern of the AUC values with different parameter values can still be well-predicted by PRStuning.

Currently, we only considered tuning parameters for PRS methods on diseases or other binary phenotypes. For quantitative phenotypes, instead of AUC, predictive $r^2$ is commonly used as an evaluation criterion of the PRS model. Extending PRStuning to evaluating predictive $r^2$ and selecting parameters on quantitative phenotypes is left as future work.

In PRStuning, we select the best-performing parameter by predicting the AUC of the PRS built under each candidate parameter value. Although AUC is the most commonly used evaluation metric for PRS on binary disease outcomes[22], it may be helpful to incorporate additional covariates such as age, sex, etc. into the AUC since they may also have an impact on disease risks[34]. Two notable variants of AUC to incorporate covariate information include covariate-specific AUC ($AUC_x$)[35] and covariate-adjusted AUC (AAUC)[36]. Similar to the definition of the ordinary AUC, $AUC_x$ is defined as the probability that the PRS of a random individual from the case group is larger than the PRS from a random individual from the control group conditioning on both individuals share the common covariate value $x$. AAUC is the weighted average of $AUC_x$ where the weight is the probability density of covariate value $x$. If the genetic risk of a disease is independent of other covariates, both $AUC_x$ and AAUC will have the same value as the ordinary AUC[34]. To estimate $AUC_x$ and AAUC, we need to estimate the conditional distribution of PRS given a covariate value, which can only be inferred with the help of individual-level data. Since we focus on using GWAS summary statistics to predict the AUC and tune parameters of PRS, we left the prediction of covariate-incorporated $AUC_x$ and AAUC based on individual-level training data as future work.

The basic assumption of PRStuning is that the training and testing datasets are homogeneous, indicating that both datasets come from the same population and therefore share the same LD matrix and expected allele frequencies among controls and cases. The same assumption is also needed for traditional PRS analyses based on an independent validation dataset to tune parameters. If the validation and testing datasets are heterogeneous, the AUC estimated from the validation dataset and the parameter selected based on the estimated results are not accurate. Without additional information about the heterogeneity between the two datasets, it will be challenging to estimate AUC and tune parameters based on training or validation

datasets. We note that some recent PRS methods have been proposed to consider multiple populations from different ancestries together, which can transfer the knowledge from the European population to other demographics with limited sample size[37–40]. In PRStuning, we currently focus on dealing with the overfitting issue when the homogeneous assumption is valid. The adjustment to the selected parameter value based on additional information of the heterogeneity will be considered in our future work. Supplementary Figures 3-13 present the performance of PRStuning when the pooled allele frequency, effect size, and LD matrix are different between training and testing datasets. The figures demonstrate that PRStuning can estimate the AUC well when heterogeneity exists in the pooled allele frequency and LD matrix. However, if the heterogeneity between training and testing data exists in the effects of changing allele frequencies between controls and cases, the AUC from PRStuning will be overestimated and unreliable.

Recent research suggests that combining all PRSs under a tuning grid using ensemble methods can improve the prediction performance[8,41–43]. In the ensemble methods, an independent validation dataset is needed to estimate the weights used for combining PRSs. In PRStuning, we estimate the AUC and select the best-performing parameters for a PRS method based on the SNP weights derived from the PRS method. If the PRS weights used in ensemble methods have already been estimated in an individual-level validation dataset, we can combine the SNP weights in each PRS and the PRS weights together to derive the ensembled SNP weights. In this situation, PRStuning can be used to predict the AUC of the PRS from the ensembled weights without another individual-level dataset. However, without an individual-level validation dataset to estimate the PRS weights used in the ensemble methods, PRStuning can not estimate the PRS weights simply based on GWAS summary statistics from the training data.

## Methods

### Notations and assumptions

Based on the additive assumption, the PRS for individual $i$ is the sum of the genotypes $g_i = (g_{i,1}, ..., g_{i,M})$ weighted by the corresponding effects $\omega = (\omega_1, ..., \omega_M)$:

$$PRS_i = \sum_{m=1}^{M} \omega_m g_{i,m}, \qquad (9)$$

where $M$ is the total number of the pre-selected SNPs used for constructing PRS. Depending on the specific PRS method, not all SNPs collected in the training GWAS data are necessarily used in PRS calculation. Please note that some PRS methods incorporate steps for selecting SNPs based on their associations with the disease. Here we define the pre-selected SNPs as the SNPs used in building the PRS model before running a selection step related to association strengths. LD clumping is an example of the selection step based on the observed association strength. Hence, we refer to the pre-selected SNPs in C+T as genome-wide SNPs collected in the training GWAS data. On the contrary, LD pruning is a selection step unrelated to the associations of SNPs with the disease. Therefore, the pre-selected SNPs in P+T are the SNPs selected after an LD pruning step. Different PRS methods have been proposed to estimate the weight vector $\omega = (\omega_1, ..., \omega_M)$ from a GWAS dataset or its summary statistics for the disease of interest. Here and after we simply use $\omega$ to denote the effects already estimated from a PRS method.

Based on disease status, we divide individuals into the case and control groups. In the following, we use subscripts $j = 0$ and $j = 1$ to denote those from the control and case groups, respectively. For example, the frequencies of the reference allele for SNP $m$ among controls and cases are denoted as $f_{0,m}$ and $f_{1,m}$, respectively. The genotype $g_{i,m}$ of SNP $m$ for an individual in the control group follows a binomial distribution $Bino(2, f_{0,m})$ with mean $\mathbb{E}[g_{0,m}] = 2f_{0,m}$ and

variance $s_{0,m}^2 := Var(g_{0,m}) = 2f_{0,m}(1 - f_{0,m})$. Similarly, we have $g_{i,m} \sim Bino(2, f_{1,m})$ if the individual $i$ is from the case group.

By the central limit theorem, PRS approximately follows a normal distribution in each group when the SNP number $M$ is adequately large. For PRS methods involving SNP selection steps unrelated to the SNPs' associations with the disease, such as P+T, $M$ varies from ~10 to ~10$K$ depending on the selection threshold. For PRS methods with genome-wide pre-selected SNPs, $M$ ranges from ~100$K$ to ~1$M$ determined by the number of SNPs genotyped or imputed in the training data. Based on the central limit theorem, the PRS variables from the two groups follow normal distributions:

$$PRS_i \sim \begin{cases} N(\eta_0, \tau_0^2) & \text{if } i \in \text{control group} \\ N(\eta_1, \tau_1^2) & \text{if } i \in \text{case group} \end{cases}, \qquad (10)$$

where

$$\eta_j = \sum_{m=1}^{M} 2\omega_m f_{j,m}, \qquad (11)$$

and

$$\tau_j^2 = \sum_{m=1}^{M} \omega_m^2 s_{j,m}^2 + 2 \sum_{m_1 < m_2} \omega_{m_1} \omega_{m_2} R_{m_1,m_2} s_{j,m_1} s_{j,m_2}, \qquad (12)$$

for $j = 0$ or 1. Here $R_{m_1,m_2}$ corresponds to the correlation between SNP $m_1$ and SNP $m_2$, which is known as the LD coefficient.

For a binary phenotype, we usually use AUC as the criterion for evaluating the prediction performance of PRS. AUC is defined as the area under the ROC curve, which can also be calculated as the probability that a random PRS from the case group is larger than a random PRS from the control group[44]. Based on this fact and the distributions of PRS, Song, etc.[22] formulated AUC as

$$AUC = \Phi(\Delta), \qquad (13)$$

where

$$\Delta := \frac{\eta_1 - \eta_0}{\sqrt{\tau_0^2 + \tau_1^2}} = \frac{2\sum_{m=1}^{M} \omega_m \delta_m}{\sqrt{\tau_0^2 + \tau_1^2}}. \qquad (14)$$

Here $\delta_m := f_{1,m} - f_{0,m}$ records the difference between the allele frequencies of the two groups for SNP $m$, and $\Phi(\cdot)$ is the cumulative density function of a standard normal distribution.

To calculate $\tau_0^2$ and $\tau_1^2$ in Eq. (13), we can directly plug in the observed values of the allele frequencies and LD coefficients into Eq. (12) since they are not directly related with the SNP effects on the disease. We can extract allele frequencies from summary statistics of the GWAS and use a genotyping dataset as the reference panel for extracting the LD information. Some large projects, such as the 1000 Genomes project[26] and the HapMap3 project[27], can be used to calculate the LD coefficients. We will provide the details of calculations in Section "Calculating LD from a reference pane".

In Eq. (13), the allele frequency differences $\delta_m$ ($m = 1, ..., M$) are critical. One may think of directly plugging in the observed allele frequencies $\hat{f}_{0,m}$ and $\hat{f}_{1,m}$ from GWAS for building the PRS model to obtain $\delta_m$. However, the allele frequency differences of SNPs that exhibit large effects tend to be overestimated, and these SNPs have larger contributions to PRS than the SNPs showing small effects, a phenomenon known as overfitting in the machine learning community[23]. Overestimating the SNP effects would lead to an inflated value of the predicted AUC and the incorrectly selected values of the parameters. Here we adopt an empirical Bayes method to reduce the influence of overfitting by shrinking the observed allele frequency

differences obtained from the summary statistics of the training GWAS.

In GWAS, we usually use the z-score calculated from the allele frequency difference test to assess the association of each SNP with the disease. Since z-scores are standardized values following a standard normal distribution $N(0,1)$ under the null hypothesis, we will use z-scores as surrogates to derive the posterior distribution of $\delta_m$. The z-score is calculated with the following formula:

$$z_m = \frac{\hat{f}_{1,m} - \hat{f}_{0,m}}{\sqrt{s_{1,m}^2/4n_1 + s_{0,m}^2/4n_0}}, \tag{15}$$

where $\hat{f}_{j,m}$ is the observed allele frequencies among controls or cases, and $s_{j,m}^2$ is the variance of genotypes in each group. We use $n_0$ and $n_1$ to respectively denote the sample sizes of controls and cases in the GWAS. To simplify the expression, we use $s_m$ to denote the denominator of the z-score, i.e.,

$$s_m := \sqrt{s_{1,m}^2/4n_1 + s_{0,m}^2/4n_0}, \tag{16}$$

and denote $s = (s_1, ..., s_M)$. We use $z$ to encode the z-scores of all the pre-selected SNPs. Based on this definition, we have $z_m|\delta_m \sim N(\delta_m/s_m, 1)$ given the allele frequency difference $\delta_m$.

Under an assumed condition that SNP $m$ is independent of other SNPs, its potential allele frequencies among controls and cases may be different. We denote the potential allele frequencies under this condition as $p_{0,m}$ and $p_{1,m}$, respectively. Note that they should be distinguished from the marginalized allele frequencies $f_{0,m}$ and $f_{1,m}$. We denote the effect of SNP $m$ as $\beta_m = p_{1,m} - p_{0,m}$. If the SNP has no effect on the disease, then $\beta_m = 0$. For the risk ones, $\beta_m \neq 0$. In the Supplementary Methods section, we further prove that $\delta = (\delta_1, ..., \delta_M)$ is actually related to the LD among the pre-selected SNPs and the underlying SNP effects $\beta = (\beta_1, ..., \beta_M)$ in terms of changing allele frequencies between two groups, i.e.,

$$delta = SRS^{-1}\beta. \tag{17}$$

We further assume that the standardized effect $\beta_m/s_m$ follows a point-normal distribution, i.e.,

$$\frac{\beta_m}{s_m} \overset{iid}{\sim} (1 - \pi)\delta_0 + \pi N(0, \sigma^2). \tag{18}$$

Here $\delta_0$ is a point mass at zero and $\pi$ represents the prior proportion of the SNPs having effects on the disease. We use $\sigma^2$ to denote the variance of $\beta_m/s_m$ in the risk SNPs. In the Supplementary Methods section, we derived the following relationship between $\sigma^2$ and the heritability ($h_l^2$) of the disease in the liability-scale:

$$\sigma^2 = \frac{N_e h_l^2}{4M\pi} \frac{\phi(\Phi^{-1}(\kappa))^2}{\kappa^2(1-\kappa)^2}, \tag{19}$$

where $N_e = \frac{4n_0n_1}{n_0+n_1}$ is the effective sample size of the GWAS, $\kappa$ is the prevalence of the disease, and $\phi$ and $\Phi$ are the probability density function and cumulative density function of the standard normal distribution $N(0,1)$, respectively.

In the following two subsections, we will demonstrate how to estimate allele frequency differences in two different scenarios by reducing the effect of overfitting based on the empirical Bayes theory.

### Estimating AUC on independent SNPs
First, we consider the situation in which the pre-selected SNPs used for constructing PRS are independent. For example, the pre-selected SNPs

in P+T are approximately independent because they are selected after an LD pruning step.

In this scenario, we have $\delta = \beta$ based on Eq. (17) and the joint distribution of z-scores follows a multivariate normal distribution with the covariance matrix equaling to the identity matrix $I_M$, i.e.,

$$z|\beta \sim N_M(S^{-1}\beta, I_M), \tag{20}$$

where $S = diag(s)$ is a diagonal matrix with diagonal elements encoding the standard errors of the observed allele frequency differences.

With the point-normal prior (18) on each entry of $\beta$, the log-likelihood of the z-scores is the summation of the log-likelihood for each individual z-score, i.e.

$$\log P(z|\pi, \sigma^2) = \sum_{m=1}^{M} \log P(z_m|\pi, \sigma^2). \tag{21}$$

With this property, we can use an EM algorithm to get estimates of $\pi$ and $\sigma^2$ by maximizing the likelihood $P(z|\pi, \sigma^2)$.

After getting estimates of parameters $\pi$ and $\sigma^2$, we can derive a closed-form solution for the posterior distribution of $\delta_m$:

$$\delta_m|z_m \sim (1 - h_m)\delta_0 + h_m N(\lambda z_m s_m, \lambda s_m^2), \tag{22}$$

where

$$h_m = \frac{\frac{\pi}{\sqrt{1+\sigma^2}}\phi(z_m/\sqrt{1+\sigma^2})}{(1-\pi)\phi(z_m) + \frac{\pi}{\sqrt{1+\sigma^2}}\phi(z_m/\sqrt{1+\sigma^2})} \text{ and } \lambda = \frac{1}{1+1/\sigma^2}. \tag{23}$$

Here $\phi(\cdot)$ is the probability density function of a standard normal distribution $N(0,1)$. Derivation details of this posterior distribution can be found in the Supplementary Methods section. With Eq. (22), we get MC samples of $\delta_m|z_m$ and plug them as the allele frequency difference in Eq. (13) for calculating the posterior distribution of AUC. The shrink estimator of $\delta_m$ in (22) reduces the effect of overfitting. Details of the EM algorithm for estimating $\pi$, $\sigma^2$, $\delta_m$, and AUC are summarized in Algorithm 4.2.

**Algorithm 1.** Estimate AUC on independent SNPs
> **Input:** z-scores $z = (z_1, ..., z_M)$
> **Output:** Estimated $\pi$, $\sigma^2$, $\delta$ and AUC
> 1: Initialize $\pi$ and $\sigma^2$;
> 2: **repeat**
> 3: **for** $m = 1, 2, ..., M$ **do**
> 4: E step:
> 5: $h_m \leftarrow \frac{\pi\phi(z_m/\sqrt{1+\sigma^2})/\sqrt{1+\sigma^2}}{(1-\pi)\phi(z_m) + \pi\phi(z_m/\sqrt{1+\sigma^2})/\sqrt{1+\sigma^2}}$
> 6: M step:
> 7: $\pi \leftarrow \frac{\sum_{m=1}^{M} h_m}{M}$
> 8: $\sigma^2 \leftarrow \frac{\sum_{m=1}^{M} h_m z_m^2}{\sum_{m=1}^{M} h_m} - 1$
> 9: **end for**
> 10: **untill** $\pi$ and $\sigma^2$ converge
> 11 **for** $m = 1, 2, ..., M$ **do**
> 12: $\delta_m \sim (1 - h_m)\delta_0 + h_m N(\frac{z_m s_m}{1+1/\sigma^2}, \frac{s_m^2}{1+1/\sigma^2})$
> 13: **end for**
> 14: $\Delta \leftarrow \frac{2\sum_{m=1}^{M} \omega_m \delta_m}{\sqrt{\tau_0^2 + \tau_1^2}}$ and AUC $\leftarrow \Phi(\Delta)$

### Estimating AUC on SNPs linked by LD
When the pre-selected SNPs are not filtered by the independence criterion, their genotypes may be correlated due to LD. We can estimate the LD matrix $R$ from a publicly available genotyping reference panel.

In this scenario, we have $\delta = SRS^{-1}\beta$ based on Eq.(17) and the conditional joint distribution of the $z$-scores given $\beta$ is

$$z|\beta \sim N(RS^{-1}\beta, R), \tag{24}$$

where $S = diag(s)$ is a diagonal matrix encoding the standard errors of observed allele frequency differences.

We used the same point-normal prior (18) on each entry of $\beta$ as we used in the independent SNP scenario. There are two unknown parameters $\pi$ and $\sigma^2$ in the prior distribution. We intend to use maximum likelihood estimation (MLE) for estimating them based on the observed $z$-scores. However, due to the extremely high number of component combinations (i.e., $2^M$), the joint likelihood of $z$-scores $P(z|\pi, \sigma^2)$ is intractable. Here we use a Gibbs-sampling-based State-Augmentation for Marginal Estimation (SAME) algorithm to get the maximizer of the likelihood in a stochastic approach[25].

Let $\gamma_m \in \{0, 1\}$ ($m = 1, ..., M$) denote whether SNP $m$ has an effect on the disease or not and $\gamma = (\gamma_1, ..., \gamma_M)$. In the SAME algorithm, instead of evaluating the original likelihood, we assess the likelihood of the augmented data $P(z, \beta, \gamma|\pi, \sigma^2)$. With flat priors on $\pi$ and $\sigma^2$, we derive a Gibbs sampler for sampling the full parameters $\beta$, $\gamma$, $\pi$ and $\sigma^2$ with the joint probability proportional to the augmented data likelihood. We leave the derivation details in the Supplementary Methods section.

By making some simple changes to the originally derived sampler, we can get another Gibbs sampler for simultaneously sampling $\pi$, $\sigma^2$ and $D$ artificial replicates of the nuisance parameters $\{\beta(d), \gamma(d)\}_{d=1}^{D}$, for whom the joint probability is proportional to

$$q_D\left(\pi, \sigma^2, \{\beta(d), \gamma(d)\}_{d=1}^{D}|z\right) \propto \prod_{d=1}^{D} P(z, \beta(d), \gamma(d)|\pi, \sigma^2). \tag{25}$$

Based on this probability, the generated replicates of $\{\beta, \gamma\}$ in the sampler are conditionally independent. With this new sampler, the marginal probability of $(\pi, \sigma^2)$ can be calculated by integrating/summing over all replicates of $\{\beta, \gamma\}$:

$$q_D(\pi, \sigma^2|z) = \int_{\beta(D)} \sum_{\gamma(D)} \cdots \int_{\beta(1)} \sum_{\gamma(1)} q_D\left(\pi, \sigma^2, \{\beta(d), \gamma(d)\}_{d=1}^{D}|z\right) d\beta(1) \ldots d\beta(D)$$

$$\propto \int_{\beta(D)} \sum_{\gamma(D)} \cdots \int_{\beta(1)} \sum_{\gamma(1)} \prod_{d=1}^{D} P(z, \beta(d), \gamma(d)|\pi, \sigma^2) d\beta(1) \ldots d\beta(D)$$

$$= \prod_{d=1}^{D} \left( \int_{\beta(d)} \sum_{\gamma(d)} P(z, \beta(d), \gamma(d)|\pi, \sigma^2) d\beta(d) \right)$$

$$= P(z|\pi, \sigma^2)^D.$$

In other words, $(\pi, \sigma^2)$ is actually sampled from $q_D(\pi, \sigma^2|z) \propto P(z|\pi, \sigma^2)^D$ in the sampler. We further denote $(\hat{\pi}, \hat{\sigma}^2) = \arg\max_{(\pi, \sigma^2)} P(z|\pi, \sigma^2)$ and $(\tilde{\pi}, \tilde{\sigma}^2)$ as another set of parameters. If we let $D$ increase to infinity, the relative probability of sampling $(\tilde{\pi}, \tilde{\sigma}^2)$ compared to sampling $(\hat{\pi}, \hat{\sigma}^2)$ will become

$$\frac{q_D\left(\tilde{\pi}, \tilde{\sigma}^2|z\right)}{q_D\left(\hat{\pi}, \hat{\sigma}^2|z\right)} = \left( \frac{P(z|\tilde{\pi}, \tilde{\sigma}^2)}{P(z|\hat{\pi}, \hat{\sigma}^2)} \right)^D \xrightarrow{D \to \infty} 0. \tag{26}$$

Therefore, the sampled $(\pi, \sigma^2)$ will converge to their maximum likelihood estimates $(\hat{\pi}, \hat{\sigma}^2)$ in the end.

Given their estimates, the Gibbs sampler in the SAME algorithm can provide MC samples of nuisance parameters $\{\beta, \gamma\}$ with probability $P(\beta, \gamma|z, \hat{\pi}, \hat{\sigma}^2)$. With them, we can also get the MC samples of $\delta = SRS^{-1}\beta$ and the corresponding AUC based on Eq. (13). The complete Gibbs-sampling-based SAME algorithm for estimating $\pi$, $\sigma^2$, $\delta_m$ and AUC is summarized in Algorithm 4.3.

**Algorithm 2.** Estimate AUC on SNPs linked by LD

**Input:** $z$-scores $z = (z_1, ..., z_M)$
**Output:** Estimated $\pi$, $\sigma^2$, $\delta$ and AUC
Initialize $\pi$, $\sigma^2$, $\gamma_m \sim Bernoulli(\pi)$ and $\beta_m \sim (1 - \gamma_m)\delta_0 + \gamma_m N(0, \sigma^2)$ for $m = 1 \ldots M$
$D \leftarrow 1$
$\lambda \leftarrow \frac{1}{1 + 1/\sigma^{-2}}$
**repeat**
 **for** $d \leftarrow 1$ to $D$ **do**
 **for** $m \leftarrow 1$ to $M$ **do**
 **If** $\gamma_m = 0$, $\beta_m \leftarrow 0$
 $\mu_m \leftarrow \lambda(z_m - \sum_{m' \neq m} \frac{R_{mm'}\beta_{m'}}{s_{m'}})$
 **If** $\gamma_m = 1$, sample $\beta_m \sim N(s_m\mu_m, \lambda s_m^2)$
 $r_m \leftarrow \pi \sqrt{\frac{\lambda}{\sigma^2}} \exp(\frac{\mu_m^2}{2\lambda})$
 $h_m \leftarrow \frac{r_m}{(1-\pi) + r_m}$
 Sample $\gamma_m \sim Bernoulli(h_m)$
 **end for**
 $\beta(d) \leftarrow \beta$ and $\gamma(d) \leftarrow \gamma$
 **end for**
Sample $\pi \sim Beta\left( \sum_{d=1}^{D} \sum_{m=1}^{M} \gamma_m(d) + D, MD - \sum_{d=1}^{D} \sum_{m=1}^{M} \gamma_m(d) + D \right)$
Sample $\sigma^{-2} \sim Gamma\left( \frac{1}{2} \sum_{d=1}^{D} \sum_{m=1}^{M} \gamma_m(d) + D, \frac{1}{2} \sum_{d=1}^{D} \sum_{m=1}^{d} \beta_m(d)^2 \gamma_m(d) \right)$
$D \leftarrow D + 1$
**until** $(\pi, \sigma^2)$ converge.

$\delta \leftarrow SRS^{-1}\beta$, $\Delta \leftarrow \frac{2\sum_{m=1}^{M} \omega_m \delta_m}{\sqrt{\tau_0^2 + \tau_1^2}}$ and AUC $\leftarrow \Phi(\Delta)$

## Calculating LD from a reference panel

Algorithm 4.3 needs users to input the LD matrix among the pre-selected SNPs. Some projects, such as the 1000 Genomes Project[26] and the HapMap 3 project[27] have released individual-level genotype data. We can use them as reference panels to extract the LD matrix. In our method, we chose the 1000 Genomes Project as our default reference panel since it has a larger sample size. Note that most PRS methods calculate weights on the SNPs genotyped in the HapMap 3 project (HM3 SNPs) because it constitutes a set of commonly used tag SNPs that are usually well-imputed in different GWAS. To extract reliable results of the LD matrix and to reduce the computational cost of Algorithm 4.3, we only included HM3 SNPs in the reference panel in our experiments.

We note that the LD coefficient between SNPs tends to decay with increasing distance between SNPs[45]. The genotypes of SNPs with a long distance are approximately independent. We use LDetect to divide the whole genome into approximately independent blocks[46]. For human genomes with European ancestry, a total of 1703 blocks are partitioned by LDetect.

Within each partitioned block, the correlation matrix among the genotypes of SNPs needs to be estimated as an input. Many methods have been proposed to estimate SNP covariance matrix[47-49], but most of them are sensitive to the structure of the covariance matrix or the distribution of the sample data. We note that the Ledoit-Wolf estimator does not depend on the assumptions of the covariance structure or the sample data distribution[49]. In our method, we first standardized genotypes in the reference panel, and then we adopted the Ledoit-Wolf estimator on the standardized genotypes to obtain the correlation matrix.

## Reporting summary

Further information on research design is available in the Nature Portfolio Reporting Summary linked to this article.

## Data availability

The 1000genomes data can be downloaded via https://www.internationalgenome.org/, and the HapMap3 data can be downloaded

via https://www.sanger.ac.uk/resources/downloads/human/hapmap3.html. The UK Biobank data are available under restricted access. Researchers can apply for access at https://www.ukbiobank.ac.uk/enable-your-research/apply-for-access. The Type 2 Diabetes GWAS summary level data available from the DIAGRAM consortium [https://diagram-consortium.org/downloads.html]. The Coronary Artery Disease GWA meta-analysis data are available from the CARDIoGRAM Consortium [http://www.cardiogramplusc4d.org/]. The Inflammatory Bowel Disease Disease GWAS summary level data are available from the IIBDGC consortia [https://www.ibdgenetics.org/]. The Breast cancer data are available from the BCA Consortium [https://bcac.ccge.medschl.cam.ac.uk/bcacdata/oncoarray/oncoarray-and-combined-summary-result/gwas-summary-results-breast-cancer-risk-2017]. We provide example data for demonstrating the usage of our method at https://github.com/lscientific/PRStuning, where the reference panel and corresponding LD matrix based on the 1000 Genomes Project can also be found. Source data are provided with this paper.

## Code availability

The codes for PRStuning are available at https://github.com/lscientific/PRStuning. Permanent repositories are available at https://doi.org/10.5281/zenodo.10119783[50].

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

## Acknowledgements

We sincerely thank CARDIoGRAM, IIBDGC, DIAGRAM, and BCA consortia for the publicly accessible GWAS summary statistics. This study makes use of data generated by the UK Biobank under Application Number 29900. A full list of the investigators who contributed to the generation of the data is available from https://www.ukbiobank.ac.uk/. Our work was supported in part by the National Institutes of Health (https://www.nih.gov/) grants R01 HG012735, R01 GM134005, and National Science Foundation (https://www.nsf.gov/funding/) grant DMS1902903, received by H.Z.. The funders had no role in study design, data collection and analysis, decision to publish, or preparation of the manuscript.

## Author contributions

W.J. designated the idea of this work and developed the main body of algorithms. L.C. performed the simulation and real data experiments and developed part of the algorithms. W.J. and L.C. wrote the manuscript. M.J.G. and H.Z. supervised the project, revised the manuscript, and commented on it.

## Competing interests

The authors declared no competing interests.
