## [Peer Review File · Nature Communications]

Tuning Parameters for Polygenic Risk Score Methods Using GWAS Summary Statistics from Training DataREVIEWER COMMENTS

Reviewer #1 (Remarks to the Author):

Report: Tuning Parameters for Polygenic Risk Score Methods Using GWAS Summary Statistics from Training Data

The main contribution of this paper is, Authors propose a new method that can utilize summary statistics, training data of PRS development techniques, to find the optimal parameters of the P+T and LDpred PRS development techniques and calculate AUCs. This new method is built on Song's method, summaryAUC (<https://doi.org/10.1093/bioinformatics/btz176>) which require only summary statistics of the validation data to calculate AUC, instead of individual level data sets. As those Authors mentioned in that paper, the summaryAUC is best used for PRSs with independent SNPs and for PRSs with $\leq 20,000$ SNPs and not suitable for PRSs integrating all common SNPs in the genome such as LD-Pred as it is computationally infeasible to adjust for the correlation for multiple millions of SNPs.

In this paper author proposed a new technique, PRStunning method, which is built on summaryAUC, uses the same training data to find the optimal parameter of techniques such P+T and LDpred. In addition, Authors calculated AUCs using the summary statistics and compare the performance of the PRStunning on P+T, C+T and LDpred using another individual level data, test data sets. The author calculated the correlation (ρ AUC) of the PRStuning-predicted AUC values with those estimated on the testing data. A high value of ρ AUC indicates that the predicted AUC using proposed method is highly correlated with the AUC on the testing data.

1. In this paper Authors tried different approaches to calculate AUC, it would be good to give additional informations such as the number of SNPs kept for each threshold of P+T, C+T etc. Are the P+T and C+T techniques used only HapMap3 based SNPs? What did the Authors mean about threshold=1 for P+T/C+T techniques. Please clarify it.

2. Would it be possible to include any confounders such as age, sex etc. as part of the AUC calculation?

3. Authors used Hapmap3 based SNPs in LDpred model and changed the proportion of risk SNPs from $\{1, 3e-1, 1e-1, 3e-2, 1e-2, 3e-3, 1e-3, 3e-4, 1e-4, 3e-5, 1e-5\}$ as the parameter in the model. Author used 80% of UKBB data for training, including generating summary statistics for PRStunning method. Later they used the remaining 20% for the testing of P+T, C+T and LDpred based on PRStuning techniques. Could Authors please clarify the following questions?

a. The author stated that they used the same point-normal prior as described in <https://doi.org/10.1038/s41467-019-09718-5> with two unknown parameters π and σ^2 in the prior distribution. Could Authors please define these parameters based on the prior used in this paper?

b. In addition to the proportion of the risk SNPs, will author consider the heritability estimates of the SNPs as well in the model?

c. How did Authors ensure that Gibbs chains are converged in the model?

d. Is there any convergence issue if the LD matrix is generated from samples which are not part of training data?

e. The author mentioned that LD radius was set to 5, what is this, please explain this value in terms SNPS, mega base pair or genetic distances etc.?

f. How does this proposed method handle the HLA region of Chromosome 6, is the chosen LD radius large enough to handle HLA region? Please clarify it?

g. In the predicted AUC values for PRStunning based on C+T with different p-value thresholds in the simulation experiment based on the UKBB data (Table 4), the AUC increases when threshold decreases. How did Authors obtain the optimal threshold for this approach? We could observe similar pattern for PRStunning based on LDpred (Table 5) the highest AUC is obtained when the parameter is 0.3. Is it an indication of the overfitting in the model or are there any convergence issue in the model? Based on Table 4 and 5, is it concluded that P+T is better than LDpred model?

4. As the summary statistics of Type 2 Diabetes, Coronary Artery Disease and Inflammatory Bowel Disease are available, could Authors generate PRS using traditional PRS development technique such LDpred2 and obtain the optimal parameters using UKBB data and compare the AUC with the LDpred based on PRStuning technique?

5. It would be great to include the computational time for different approaches Authors have tried in this paper.

6. PRS-CSx and LDpred2 provide LD matrix as a part of their software based on 1000G or UKBB data sets. It would be good to include the LD matrix as a part of the software so that others can re-use it. Are the Authors suggested to use only genotyped data for calculating LD matrix as the bed file may convert dosage to hard calls? What would be the estimated time to calculate LD matrix for HapMap3 based SNPs for 1000 samples? If we include more samples in the LD matrix calculation, would it affect the predictive performances? How did the software generate LD blocks or the pre-define LD blocks are available as part of the software? Is the complete code for generating LD matrix are given as a part of the software?

7. Author mentioned that they have applied the proposed framework in European ancestral group. Would it be possible to include non-European population in the training datasets?

Reviewer #2 (Remarks to the Author):

I have attached my review comments as a word document in the attachment since there are some math formulas in it.

Dr. Jiang and colleagues present a novel approach, PRStuning, to evaluate PRS performance solely relying on GWAS summary statistics from the training dataset. This method extends upon the SummaryAUC approach proposed by Song et al. in 2019, which employs GWAS summary statistics from an external testing dataset to assess the AUC of PRS. The paper further refines the SummaryAUC approach by introducing a 'spike and slab' prior and utilizing Empirical Bayes (EB) theory to shrink effect sizes. This approach offers two significant advantages:

- There is no requirement for individual or GWAS summary statistics from an independent testing data for PRS development.
- The original testing data can be incorporated into the training data, thereby enlarging the sample size for PRS prediction.

In my view, these innovations represent an important contribution to the field of PRS. I have a few comments and questions:

Q1: Regarding Equation 5 on page 2 of the Supplementary Notes, the authors assert that “Due to the independence among pre-selected SNPs, the allele frequency difference of SNP m is equal the risk effect size, i.e., $\delta_m = \beta_m$ ”. This claim is not immediately intuitive. For example, suppose there is only one variant G , with allele frequency f_0 in the controls. Let $\Pr(Y = 1|G) = \text{logit}^{-1}(G\beta)$. Following the Bayes rule, $\Pr(G|Y = 1) = \frac{\text{logit}^{-1}(G\beta)\Pr(G)}{\sum_{G=0}^2 \text{logit}^{-1}(G\beta)\Pr(G)}$. Since G follows a binomial distribution with $B(2, f_0)$ in the control group, the allele frequency in cases is

$\sqrt{\frac{\text{logit}^{-1}(2\beta)f_0^2}{\sum_{G=0}^2 \text{logit}^{-1}(G\beta)\Pr(G)}}$. Can you give some additional explanations for $\delta_m = \beta_m$?

Q2: On Supplementary Note page 3, I found the following section unclear. Specifically, the difference between p_{A0} and f_{A0} are unclear. Additionally, p_A and p_a are not well defined. Could you please elaborate on these elements?

where $s_A = \sqrt{\frac{\hat{f}_{A1}(1-\hat{f}_{A1})}{2n_1} + \frac{\hat{f}_{A0}(1-\hat{f}_{A0})}{2n_0}}$ is the standard error of $\hat{f}_1 - \hat{f}_0$ based on the variance formula of a binomial distribution. Here we assume the underlying real allele frequencies without the influence of other SNPs among cases and controls are p_{A1} and p_{A0} , respectively. we define the true effect of SNP A as $\beta_A = p_{A1} - p_{A0}$. We use f_{A1} and f_{A0} denoting the expectations of observed allele frequencies among cases and controls. We have

$$s_A \approx \sqrt{\frac{p_{A1}p_{a1}}{2n_1} + \frac{p_{A0}p_{a0}}{2n_0}} \approx \sqrt{\left(\frac{1}{2n_1} + \frac{1}{2n_0}\right)p_A p_a}$$

Q3: The simulation results displayed consistent outcomes between PRStuning and testing data for three PRS methods: P + T, C + T, and LDpred. This was insightful. However, could you also assess the performance of PRS-CS, another commonly utilized PRS method? Furthermore,

LDpred2, an updated version of LDpred, has seen frequent use in recent applications. It may be more appropriate to utilize LDpred2 as the benchmark method, rather than LDpred.

Q4: On page 13 of the Results section, I suggest providing the precise number of independent individuals of European ancestry included in the UK Biobank (UKB) dataset.

Q5: C + T methods also rely on approximately independent SNPs to construct the PRS. Why did the authors opt for a Gibbs sampling-based SAME algorithm for C + T in the analyses, rather than an EM algorithm-based approach?

Q6: In Figure 4, there appears to be a high consistency between PRStuning and testing data results for P + T. Yet, the correlation ρ_{AUC} seems low (0.383-0.783) as shown in Table 7. Conversely, LDpred results seem less consistent in Figure 4, but exhibit a high correlation ρ_{AUC} (0.969-0.987) in Table 7. Could you clarify this discrepancy?

Q7: Including additional real data results could help further validate the accuracy of PRStuning. For instance, the Breast Cancer Association Consortium has released GWAS summary statistics. Including breast cancer in your analyses as an additional example could be beneficial.

Q8: Traditional PRS analyses often divide data into three independent sets: training, testing, and validation. The prediction could still be overfitted toward the testing data if only the best performance on the testing dataset is reported. While PRStuning can estimate the AUC for the testing dataset based solely on the GWAS summary statistics of the training dataset, how do you avoid potential overfitting when reporting the best prediction PRS after evaluating all the tuning grids?

Q9: Recent PRS literature suggests that instead of selecting the best performing PRS based on the testing dataset, combining all PRSs under a tuning grid using ensemble approaches generally yields superior prediction performance (as illustrated in the following studies:

<https://pubmed.ncbi.nlm.nih.gov/31761295/>

<https://www.biorxiv.org/content/10.1101/2022.03.24.485519v5.full.pdf>

<https://www.biorxiv.org/content/10.1101/2023.03.15.532652v1.full.pdf>

<https://www.biorxiv.org/content/10.1101/2023.04.12.536510v1.full.pdf>

Can PRStuning incorporate an ensemble step into its framework? If not, a discussion on this topic would be useful.

Q10. This paper presents convincing evidence, through extensive simulations and real data analyses, that the proposed PRStuning approach can produce consistent results with an external testing dataset. However, additional theoretical intuition would be valuable. Specifically, it's somewhat counterintuitive why incorporating shrinkage via the Bayesian framework can entirely eliminate potential overfitting.

Reviewer #3 (Remarks to the Author):

Jiang et al. proposed PRStuning to automatically tune hyperparameters for PRS calculation using summary statistics from the training data. There are many statistical and machine learning methods have been developed to calculate PRS, but all of them require independent validation sets with raw genotype data to finetune the hyperparameter. I believe PRStuning is timely and would draw much attention in the field. Generally speaking, the manuscript is well prepared and with clear structures, but it lacks evidence to prove it outperforms the other methods.

Major:

1. The authors should discuss and benchmark the methods solving the same problem in the manuscript, such as PUMAS <https://doi.org/10.1186/s13059-021-02479-9>. It is not clear why PRStuning outperforms PUMAS which also uses summary statistics for model selection.
2. Regarding the manuscript, it is observed that PRStuning has only been tested on the UK Biobank dataset. The manuscript does not clarify the performance of PRStuning in cross-population studies and how to address differences in LD patterns. The authors should address the following two scenarios: 1) The summary statistics used for training data are from a multi-ancestry meta-analysis and the testing data is from different populations, and 2) The summary statistics used for training data are from Caucasian populations and the testing data is from Asia, Africa, etc.
3. Follow-up the last question, PRStuning should also be applied to the methods specifically designed for cross-population research, such as PRS-CSx, TL-PRS, SDPRX.
4. In Figure 1, please interpret why the difference between unadjusted AUC and PRStuning is much larger when lenient p-value thresholds are applied. Does it mean more inflation would be observed by involving more SNPs?
5. The authors replicated 50 times in simulating independent SNPs while replicated only 20 times in simulating correlated SNPs. Is there a reason for reducing the number of replicates?
6. Adding an additional metric to measure the proportion of shared cases between the best-performing hyperparameter from PRStuning and the testing data would be helpful. This metric would provide more detailed information for comparing PRStuning with the benchmark.
8. Please justify the criteria for determining the number of cases and controls in the simulation.
9. Can the authors show more details of $\delta = SRS^{-1}\beta$ in the section 4.3?
10. Song et al. (PMID: 30911754) pointed out that SummaryAUC was not suitable for PRS models including all common SNPs, e.g. LDpred. As PRStuning is built on the top of SummaryAUC and this fact was not fully discussed in the manuscript.
11. Can the authors provide all required files of an example to run PRStuning on Github?

Minor:

1. In line 74, should it be 'and four normal distributions' rather than 'and three normal distributions'?

2. It seems that 'parameters' and 'hyperparameters' are used interchangeably in the manuscript, which should be avoided. For example, in line 88, 'evaluate different parameter values' is supposed to be 'evaluate different hyperparameter values'. In addition, in line 91, 'just for tuning parameters' is supposed to be 'just for tuning hyperparameters'.
3. In line 416, 'PRS methods have proven useful' -> 'PRS methods have been proven useful'.
4. In line 474, 'For example, the frequency' -> 'For example, the frequencies'.
6. When testing the codes on Github, I found that I was required to install rpy2 package and R, but this was not pointed out in the dependency part of Github. Such information should be included.
7. In the dependency part of Github, 'pysnpools scikit-learn Pandas arspy' is not separated by ';'.
8. The data and code availability session should be added to the manuscript.

Tuning Parameters for Polygenic Risk Score Methods Using GWAS Summary Statistics from Training Data

Reviewer #1:

The main contribution of this paper is, Authors propose a new method that can utilize summary statistics, training data of PRS development techniques, to find the optimal parameters of the P+T and LDpred PRS development techniques and calculate AUCs. This new method is built on Song's method, summaryAUC (<https://doi.org/10.1093/bioinformatics/btz176>) which require only summary statistics of the validation data to calculate AUC, instead of individual level data sets. As those Authors mentioned in that paper, the summaryAUC is best used for PRSs with independent SNPs and for PRSs with $\leq 20,000$ SNPs and not suitable for PRSs integrating all common SNPs in the genome such as LD-Pred as it is computationally infeasible to adjust for the correlation for multiple millions of SNPs.

In this paper author proposed a new technique, PRStunning method, which is built on summaryAUC, uses the same training data to find the optimal parameter of techniques such P+T and LDpred. In addition, Authors calculated AUCs using the summary statistics and compare the performance of the PRStunning on P+T, C+T and LDpred using another individual level data, test data sets. The author calculated the correlation (ρ AUC) of the PRStuning-predicted AUC values with those estimated on the testing data. A high value of ρ AUC indicates that the predicted AUC using proposed method is highly correlated with the AUC on the testing data.

Response: We appreciate the reviewer for summarizing the main contributions of our work. Please find our point-by-point responses below for addressing your concerns. Hope these responses are satisfactory.

1. In this paper Authors tried different approaches to calculate AUC, it would be good to give additional informations such as the number of SNPs kept for each threshold of P+T, C+T etc. Are the P+T and C+T techniques used only HapMap3 based SNPs? What did the Authors mean about threshold=1 for P+T/C+T techniques. Please clarify it.

Response: Thank you for your suggestions and questions. We summarized the numbers of SNPs kept for each threshold of P+T and C+T in Supplementary Table 4.

Since we used the UKBB dataset to evaluate the performance of PRStuning, we only considered SNPs overlapped between GWAS summary statistics and the UKBB dataset. In the UKBB dataset, since only SNPs presented in the HapMap 3 project (HM3 SNPs) were used in the reference panel for reliable LD estimation and computation efficiency, we focused on the SNPs in HM3. This resulted in a total of 1,027,699 HM3 SNPs and 272,751 individuals passing the quality control criteria. The specific numbers of the overlapping SNPs for each disease are summarized in Table 7. To make a fair comparison, for all PRS methods considered, we only incorporated the SNPs overlapped between GWAS summary statistics and the testing data in the analyses. That is, we

only used HM3 SNPs for constructing PRS in P+T and C+T. We clarified this point in Section 2.3, Lines 452-458, Page 16.

In P+T, p-value threshold=1 means that no further filtering step based on p-values was utilized on pre-selected approximately independent SNPs after LD pruning. In C+T, p-value threshold=1 means that we conducted LD clumping on genome-wide SNPs without filtering based on p-values. We clarified these in Section 2.2, Lines 238-241, Page 7.

2. Would it be possible to include any confounders such as age, sex etc. as part of the AUC calculation?

Response: *Thank you for your question. In PRStuning, we selected the best-performing parameter by predicting the AUC of the PRS built under each candidate parameter value. Although AUC is the most commonly used evaluation metric for PRS on binary disease outcomes [1], as suggested by the reviewer, we may incorporate additional covariates such as age, sex, etc. into the AUC since they may also be informative on disease risks [2]. Two notable variants of AUC to incorporate covariate information include covariate-specific AUC (AUC_x) [3] and covariate-adjusted AUC (AAUC) [4]. Similar to the definition of the ordinary AUC, AUC_x is defined as the probability that the PRS of a random individual from the case group is larger than the PRS from a random individual from the control group conditioning on both individuals sharing a common covariate value x . AAUC is the weighted average of AUC_x where the weight is the probability density of covariate value x . If the genetic risk of a disease is independent of other covariates, both AUC_x and AAUC will have the same value as the ordinary AUC [2]. To estimate AUC_x and AAUC, we need to estimate the conditional distribution of PRS given a covariate value, which can only be inferred with the help of individual-level data. Since we focus on using GWAS summary statistics to predict the AUC and tune parameters of PRS, we left the prediction of covariate-incorporated AUC_x and AAUC based on individual-level training data as future work.*

The above discussion has been added into Section 3, Lines 581-599, Pages 20-21 of the manuscript.

3. Authors used Hapmap3 based SNPs in LDpred model and changed the proportion of risk SNPs from {1, 3e-1, 1e-1, 3e-2, 1e-2, 3e-3, 1e-3, 3e-4, 1e-4, 3e-5, 1e-5} as the parameter in the model. Author used 80% of UKBB data for training, including generating summary statistics for PRStuning method. Later they used the remaining 20% for the testing of P+T, C+T and LDpred based on PRStuning techniques. Could Authors please clarify the following questions?

a. The author stated that they used the same point-normal prior as described in <https://doi.org/10.1038/s41467-019-09718-5> with two unknown parameters π and σ^2 in the prior distribution. Could Authors please define these parameters based on the prior used in this paper?

Response: Thank you for your suggestion. In Supplementary Note Section 3, we have derived the relationship between the prior distribution of the standardized effect and the heritability (h_i^2) of the disease in the liability-scale, i.e.,

$$\sigma^2 = \frac{N_e h_i^2 \phi(\Phi^{-1}(\kappa))^2}{4M\pi \kappa^2(1-\kappa)^2},$$

where $N_e = \frac{4n_0n_1}{n_0+n_1}$ is the effective sample size of the GWAS, κ is the prevalence of the disease, and ϕ and Φ are the probability density function and cumulative density function of the standard normal distribution $N(0,1)$, respectively.

We have added this result into Section 4.1, Lines 733-738, Page 25 of the manuscript.

b. In addition to the proportion of the risk SNPs, will author consider the heritability estimates of the SNPs as well in the model?

Response: Thank you for your question. As we described in the response of Comment 3a, the variance (σ^2) of the point-normal prior distribution for the standardized effect is related with the liability-scale heritability of the disease. In PRStuning, instead of estimating heritability, we estimate σ^2 since it directly connects with the adjustment of the allele frequency difference δ_m ($m = 1, \dots, M$).

c. How did Authors ensure that Gibbs chains are converged in the model?

Response: Thank you for your question and we agree that convergence is very important to the iteration-based algorithm. For both the EM-algorithm used for independent pre-selected SNPs and the SAME algorithm used for the preselected SNPs with LD, we use the relative difference between the estimated parameters in two consecutive iterations as the criterion to determine whether the algorithm is converged. That is, if the following condition is satisfied,

$$\frac{|\pi^D - \pi^{D-1}|}{\pi^{D-1}} \leq 0.01 \text{ and } \frac{|\sigma^{2,D} - \sigma^{2,D-1}|}{\sigma^{2,D-1}} \leq 0.01,$$

we regard the algorithm as the converged state and output the estimation results. In the above criterion, π^{D-1} and $\sigma^{2,D-1}$ are the estimated values of π and σ^2 in the (D-1)-th iteration, and π^D and $\sigma^{2,D}$ are the estimated values in the D-th iteration.

In particular, the SAME algorithm relies on the Gibbs sampler, which needs burn-in iterations to reach the equilibrium distribution of the Monte-Carlo Markov Chain (MCMC). There are D MCMC iterations executed for the D-th iteration. The total number of MCMC iterations in all D iterations is $\frac{D(D+1)}{2}$. We regard the first 34 iterations and an additional 200 MCMC iterations (795 MCMC iterations in total) as the burn-in process. This practice is also recommended in the original paper of the SAME algorithm [5]. Once the Gibbs chain reached equilibrium, the difference of the estimated parameters in two consecutive iterations will be small. The above convergency criterion works well to assess the convergence of the Gibbs chain.

d. Is there any convergence issue if the LD matrix is generated from samples which are not part of training data?

Response: *In both simulation experiments based on genotype data from the UK Biobank (UKBB) and the real data applications, we used the genotype data from the 1000 Genomes Project as the reference panel to estimate the LD matrix. The individuals from the 1000 Genomes Project are not included in the UKBB and the GWAS providing summary statistics in the real data applications. In all scenarios, the SAME algorithm converged within 100 iterations (5,050 MCMC iterations) based on the criterion mentioned in the response of Comment 3c.*

e. The author mentioned that LD radius was set to 5, what is this, please explain this value in terms SNPS, mega base pair or genetic distances etc.?

Response: *We are sorry for the confusion made in the previous version of the manuscript. In LDpred, we need to specify another parameter named LD radius, which is the number of SNPs on each side of a given SNP for computing pairwise LD.*

In our simulation experiments with the AR(1) auto-regressive LD structure, the LD radius used in LDpred was set to 5, indicating that the SNPs used for computing LD have pairwise correlations above $0.2^5 \approx 3 \times 10^{-4}$ based on the AR(1) LD structure with auto-regressive coefficient 0.2.

In the simulation experiments based on the UKBB genotype data, the LD radius specified in LDpred was set to $\frac{M}{3000} \approx 343$, which is the default practice suggested by LDpred and corresponds to a 2Mb LD window on average in the human genome [6].

In real data applications, the LD radius in LDpred was set to $M/3000$, where M is the number of the overlapping SNPs among the GWAS summary statistics, UKBB, 1000 Genomes Project and HapMap 3 Project, which is presented in the last column of Table 7.

We added those clarifications in Section 2.2.2: Lines 344-348, Page 10-11, Section 2.2.2: Lines 418-421, Page 15, and Section 2.3: Lines 462-463, Page 16.

f. How does this proposed method handle the HLA region of Chromosome 6, is the chosen LD radius large enough to handle HLA region? Please clarify it?

Response: *Thank you for your question. The major histocompatibility complex (MHC) region explains a large proportion of heritability of immune-related disorders, including it into the PRS can improve the prediction accuracy of those disorders [6]. Therefore, we include the HLA region of Chromosome 6 for deriving the weights for PRS calculation and tuning parameters in PRStuning.*

As we described in the response of Comment 3e, the LD radius is the parameter that needs to be specified in LDpred for computing pairwise LD. We specified the LD radius to $\frac{M}{3000}$, which is the

default setting and corresponds to a 2Mb LD window on average in the human genome. Although the total length of the MHC region is around 4Mb, it can be split into three classes with around 2Mb, 1Mb and 1Mb lengths for HLA class I, II and III, respectively [7]. The LD radius specified in LDpred is large enough to cover each class of the HLA region.

In PRStuning, we use LDetect [8] to split the whole genome into 1,703 approximately independent blocks, and calculate LD matrix within each block. This is a common practice in PRS methods, such as PRS-CS [9] and SDPR [10]. LDetect identifies the breakpoints of blocks such that the overall between-block correlations are minimized. This breakpoint identification mechanism allows us to consider the LD structure within each block only. For HLA region (chr6:28,477,797-33,448,354), there are five approximately independent blocks partitioned by LDetect in total. The starting and ending positions of these five blocks on Chromosome 6 are consecutively 28917608, 29737971; 29737971, 30798168; 30798168, 31571218; 31571218, 32682664; 32682664, 33236497.

g. In the predicted AUC values for PRStuning based on C+T with different p-value thresholds in the simulation experiment based on the UKBB data (Table 4), the AUC increases when threshold decreases. How did Authors obtain the optimal threshold for this approach? We could observe similar pattern for PRStuning based on LDpred (Table 5) the highest AUC is obtained when the parameter is 0.3. Is it an indication of the overfitting in the model or are there any convergence issue in the model? Based on Table 4 and 5, is it concluded that P+T is better than LDpred model?

Response: Thank you for your insightful observation and questions. In the simulation experiments based on the UKBB genotype data, the AUC indeed increased for C+T when the p-value threshold decreased. In this situation, the optimal threshold was selected at the most stringent threshold among specified candidate thresholds, i.e., $5e-8$. This result suggests that a sparse model built from C+T has better prediction performance than a dense model, which is consistent with the specified setting of the experiments ($\pi = 0.1\%$ is used for generating effect sizes of SNPs).

For the AUC results of LDpred in the simulation experiments based on the UKBB data, we observed the dramatic decrease in the prediction performance when the specified proportion of risk SNPs was dropped from $1e-1$ to $3e-2$ and the best performance was achieved at 0.3. There is indeed a convergence issue in the LDpred model when the specified proportion is very small. The convergence issue of the LDpred leads to an unexpected phenomenon that the prediction performance decreases when sample size of the training data increases. This unexpected phenomenon has also been observed in the original papers of PRS-CS [9] and SDPR [10]. Although there is a convergence issue in LDpred, it is worth noting that PRStuning was able to detect the dramatic decrease of the prediction performance due to the issue. This further suggests the accuracy in AUC prediction and effectiveness in parameter tuning using PRStuning.

In this simulation experiment based on the UKBB data, the best performance achieved by C+T was higher than that achieved by LDpred. The same phenomenon is also observed in the real

data results of Crohn's disease and Parkinson disease presented in [11].

4. As the summary statistics of Type 2 Diabetes, Coronary Artery Disease and Inflammatory Bowel Disease are available, could Authors generate PRS using traditional PRS development technique such LDpred2 and obtain the optimal parameters using UKBB data and compare the AUC with the LDpred based on PRStuning technique?

Response: *Thank you for your suggestion, we have added both simulation and empirical experimental results using LDpred2 as PRS construction method. You can find the simulation results based on simulated genotypes in Supplementary Figure 1, the simulation results based on UKBB genotypes in Table 6, and the real data application results in Figure 4 and Table 8.*

5. It would be great to include the computational time for different approaches Authors have tried in this paper.

Response: *Thank you for your suggestion. The computational time for different methods to derive PRS and our method to tune parameters of PRS methods is summarized in Supplementary Table 3.*

6. PRS-CSx and LDpred2 provide LD matrix as a part of their software based on 1000G or UKBB data sets. It would be good to include the LD matrix as a part of the software so that others can re-use it. Are the Authors suggested to use only genotyped data for calculating LD matrix as the bed file may convert dosage to hard calls? What would be the estimated time to calculate LD matrix for HapMap3 based SNPs for 1000 samples? If we include more samples in the LD matrix calculation, would it affect the predictive performances? How did the software generate LD blocks or the pre-define LD blocks are available as part of the software? Is the complete code for generating LD matrix are given as a part of the software?

Response: *Thank you for your suggestion and questions. We have provided the LD matrix, the LD blocks defined by LDetect, and the script to calculate the LD matrix based on genotype data on the GitHub (<https://github.com/lscientific/PRStuning>). The time to calculate LD matrix for HM3 SNPs on samples from the 1000 Genomes Project is 25min after parallelized on 32 Intel Xeon 6346 cores.*

In our script, we only accept the PLINK binary format (BED file) as input to calculate LD matrix. Therefore, the dosage needs to be converted to hard calls. This is not an issue for most publicly available reference panels, such as the 1000 Genomes Project and HapMap 3 Project, because the genotypes are called from high-coverage whole genome sequencing or genome-wide SNP array.

We conducted simulation experiments to compare the performance of PRStuning based on the LD matrix estimated from the reference panel with the performance based on ground truth LD matrix. The comparison results using C+T, LDpred and LDpred2 to construct PRS can be found in Supplementary Figures 14-16, respectively. From the figures, we observe that the performance

of PRStuning based on the LD matrix estimated from 1,000 individuals are almost the same with the performance based on the ground truth LD matrix. There may be little performance improvement by including more samples into the LD matrix calculation.

7. Author mentioned that they have applied the proposed framework in European ancestral group. Would it be possible to include non-European population in the training datasets?

Response: *Thank you for your question. The basic assumption of PRStuning is that the training and testing datasets are homogeneous, indicating that both datasets come from the same population and therefore share the same LD matrix and expected allele frequencies among controls and cases. If the homogeneous assumption is satisfied, e.g., the training and testing data come from the same non-European population and the LD matrix is also estimated based on a reference panel from this population, PRStuning can also be used to predict AUC and tune parameters for PRS methods.*

The homogeneous assumption is also needed for traditional PRS analyses based on an independent validation dataset to tune parameters. If the validation and testing datasets are heterogeneous, the AUC estimated from the validation dataset and the parameter selected based on the estimated results are not accurate. Without additional information about the heterogeneity between the two datasets, it will be challenging to estimate AUC and tune parameters based on training or validation datasets. In PRStuning, we currently focus on dealing with the overfitting issue when the homogeneous assumption is valid. The adjustment to the selected parameter value based on additional information of the heterogeneity is left as future work. Supplementary Figures 2-12 present the performance of PRStuning when the pooled allele frequency, effect size and LD matrix are different between training and testing datasets. The figures demonstrate that PRStuning can estimate the AUC well when heterogeneity exists in the pooled allele frequency, and LD matrix. However, if heterogeneity between training and testing data exists in the effects of changing allele frequencies between controls and cases, the AUC from PRStuning will be overestimated and unreliable.

We have added this clarification in Section 3, Lines 500-622, Page 21.

References

1. Song, L., Liu, A., Shi, J. etc., 2019. SummaryAUC: a tool for evaluating the performance of polygenic risk prediction models in validation datasets with only summary level statistics. *Bioinformatics*, 35(20), pp.4038-4044.
2. Pardo-Fernández, J.C., Rodríguez-Alvarez, M.X. and Van Keilegom, I., 2014. A review on ROC curves in the presence of covariates. *Revstat-Statistical Journal*, 12(1), pp.21-41.
3. Dodd, L.E. and Pepe, M.S., 2003. Semiparametric regression for the area under the receiver operating characteristic curve. *Journal of the American Statistical Association*, 98(462), pp.409-417.
4. Janes, H. and Pepe, M.S., 2009. Adjusting for covariate effects on classification accuracy using the covariate-adjusted receiver operating characteristic curve. *Biometrika*, 96(2), pp.371-382.

5. Doucet, A., Godsill, S.J. and Robert, C.P., 2002. Marginal maximum a posteriori estimation using Markov chain Monte Carlo. *Statistics and Computing*, 12(1), pp.77-84.
6. Vilhjálmsson, B.J., Yang, J., Finucane, H.K., Gusev, A., Lindström, S., Ripke, S., Genovese, G., Loh, P.R., Bhatia, G., Do, R. and Hayeck, T., 2015. Modeling linkage disequilibrium increases accuracy of polygenic risk scores. *The American Journal of Human Genetics*, 97(4), pp.576-592.
7. Mizuki, N. and Kimura, M., 1996. Gene structure of the human MHC region. *Nihon Rinsho. Japanese Journal of Clinical Medicine*, 54(6), pp.1705-1717.
8. Berisa, T. and Pickrell, J.K., 2016. Approximately independent linkage disequilibrium blocks in human populations. *Bioinformatics*, 32(2), p.283.
9. Ge, T., Chen, C.Y., Ni, Y., Feng, Y.C.A. and Smoller, J.W., 2019. Polygenic prediction via Bayesian regression and continuous shrinkage priors. *Nature Communications*, 10(1), p.1776.
10. Zhou, G. and Zhao, H., 2021. A fast and robust Bayesian nonparametric method for prediction of complex traits using summary statistics. *PLoS Genetics*, 17(7), p.e1009697.
11. Song, S., Jiang, W., Hou, L. and Zhao, H., 2020. Leveraging effect size distributions to improve polygenic risk scores derived from summary statistics of genome-wide association studies. *PLoS Computational Biology*, 16(2), p.e1007565.

Reviewer #2:

Dr. Jiang and colleagues present a novel approach, PRStuning, to evaluate PRS performance solely relying on GWAS summary statistics from the training dataset. This method extends upon the SummaryAUC approach proposed by Song et al. in 2019, which employs GWAS summary statistics from an external testing dataset to assess the AUC of PRS. The paper further refines the SummaryAUC approach by introducing a 'spike and slab' prior and utilizing Empirical Bayes (EB) theory to shrink effect sizes. This approach offers two significant advantages:

- There is no requirement for individual or GWAS summary statistics from an independent testing data for PRS development.
- The original testing data can be incorporated into the training data, thereby enlarging the sample size for PRS prediction.

In my view, these innovations represent an important contribution to the field of PRS. I have a few comments and questions:

Response: We appreciate the reviewer for summarizing the advantages and contributions of our method. Please find our point-by-point responses to address your concerns below. Hope these responses are satisfactory.

Q1: Regarding Equation 5 on page 2 of the Supplementary Notes, the authors assert that “Due to the independence among pre-selected SNPs, the allele frequency difference of SNP m is equal the risk effect size, i.e., $\delta_m = \beta_m$ ”. This claim is not immediately intuitive. For example, suppose there is only one variant G , with allele frequency f_0 in the controls. Let $Pr(Y = 1|G) = \text{logit}^{-1}(G\beta)$. Following the Bayes rule, $Pr(G|Y = 1) = \frac{\text{logit}^{-1}(G\beta)\text{Pr}(G)}{\sum_{G=0}^2 \text{logit}^{-1}(G\beta)\text{Pr}(G)}$. Since G follows a binomial distribution with $B(2, f_0)$ in the control group, the allele frequency in cases

$\frac{\text{logit}^{-1}(G\beta)\text{Pr}(G)}{\sum_{G=0}^2 \text{logit}^{-1}(G\beta)\text{Pr}(G)}$. Can you give some additional explanations for $\delta_m = \beta_m$?

Response: Sorry for the confusion made to the claim due to different definitions of effect sizes. In the example presented by the reviewer, the definition of β_m is log-odds ratio. In our manuscript, β_m is the underlying effect of SNP m in terms of changing allele frequencies between controls and cases, which is different from the log-odds-ratio. Let us denote the potential allele frequencies among controls and cases under an assumed condition that SNP m is independent of other SNPs as $p_{0,m}$ and $p_{1,m}$, respectively. We have $\beta_m = p_{1,m} - p_{0,m}$. Note that $f_{j,m}$ is the allele frequency of SNP m marginalizing over other SNPs, which is different with $p_{j,m}$ ($j=0,1$). Let $\beta = (\beta_1, \dots, \beta_M)$. In the Supplementary Note, we further demonstrate that the marginalized allele frequency difference $\delta = (\delta_1, \dots, \delta_M)$ is related to the LD pattern among the pre-selected SNPs and β , i.e.,

$$\delta = SRS^{-1}\beta,$$

where S is a diagonal matrix with the m -th diagonal element equal to s_m , and R is the LD coefficient matrix. With this equality, when pre-selected SNPs are independent, we have $R = I$ and $\delta_m = \beta_m$.

We have added those clarifications in Section 4.1: Lines 718-727, Page 24.

Q2: On Supplementary Note page 3, I found the following section unclear. Specifically, the difference between p_{A0} and f_{A0} are unclear. Additionally, p_A and p_a are not well defined. Could you please elaborate on these elements?

where $s_A = \sqrt{\frac{\hat{f}_{A1}(1-\hat{f}_{A1})}{2n_1} + \frac{\hat{f}_{A0}(1-\hat{f}_{A0})}{2n_0}}$ is the standard error of $\hat{f}_1 - \hat{f}_0$ based on the variance formula of a binomial distribution. Here we assume the underlying real allele frequencies without the influence of other SNPs among cases and controls are p_{A1} and p_{A0} , respectively. we define the true effect of SNP A as $\beta_A = p_{A1} - p_{A0}$. We use f_{A1} and f_{A0} denoting the expectations of observed allele frequencies among cases and controls. We have

$$s_A \approx \sqrt{\frac{p_{A1}p_{a1}}{2n_1} + \frac{p_{A0}p_{a0}}{2n_0}} \approx \sqrt{\left(\frac{1}{2n_1} + \frac{1}{2n_0}\right)p_A p_a}$$

Response: Sorry for not defining the notations clearly. We have added the following explanation into the Supplementary Note:

“Here we assume that the potential allele frequencies among cases and controls under an assumed condition that the SNP is independent of others are p_{A1} and p_{A0} , respectively. We define the true effect of SNP A as $\beta_A = p_{A1} - p_{A0}$. We use f_{A1} and f_{A0} to denote the expectations of the observed allele frequencies among cases and controls, which are the allele frequencies marginalizing over other SNPs. We have

$$s_A \approx \sqrt{\frac{f_{A1}f_{a1}}{2n_1} + \frac{f_{A0}f_{a0}}{2n_0}} \approx \sqrt{\left(\frac{1}{2n_1} + \frac{1}{2n_0}\right) f_A f_a}$$

where $f_A = \frac{n_0 f_{A0} + n_1 f_{A1}}{n_0 + n_1}$ and $f_a = \frac{n_0 f_{a0} + n_1 f_{a1}}{n_0 + n_1}$ are the pooled allele frequencies of alleles A and a among all samples under expectation.”

The table in the Supplementary Note summarizes the notations we used in the proof.

Q3: The simulation results displayed consistent outcomes between PRStuning and testing data for three PRS methods: P+T, C+T, and LDpred. This was insightful. However, could you also assess the performance of PRS-CS, another commonly utilized PRS method? Furthermore, LDpred2, an updated version of LDpred, has seen frequent use in recent applications. It may be more appropriate to utilize LDpred2 as the benchmark method, rather than LDpred.

Response: Thank you for your suggestion. We note that PRS-CS can automatically select its global shrinkage parameter by default and in practice, recent benchmarking study showed that

there was no significant difference between prediction accuracy using the optimal parameter versus the parameter selected by itself [1]. We believe it is worth investigating the performance of predicting AUC and tuning parameter for LDpred2, whose prediction performance depends more on the tuned parameter [2]. In our revised manuscript, we have added the simulation and real data results for LDpred2 (Supplementary Figure 1 for the simulation results based on simulated genotypes, Table 6 for the simulation results based on UKBB genotypes, Figure 4 and Table 8 for the real data application results). The experimental results demonstrate that PRStuning can predict the AUC and select the best-performing parameter well for LDpred2.

Q4: On page 13 of the Results section, I suggest providing the precise number of independent individuals of European ancestry included in the UK Biobank (UKB) dataset.

Response: Thank you for your suggestion. We only incorporated independent European-ancestry individuals and HM3 SNPs in the UKBB dataset, resulting in 272,751 individuals and 1,027,699 HM3 SNPs. We have added the number of independent European-ancestry individuals in the UKBB into Section 2.3, Lines 452-458, Page 16.

Q5: C + T methods also rely on approximately independent SNPs to construct the PRS. Why did the authors opt for a Gibbs sampling-based SAME algorithm for C + T in the analyses, rather than an EM algorithm-based approach?

Response: We agree that C+T method is also based on approximately independent SNPs to construct PRS. However, the clumping step in C+T incorporates the SNP selection step based on the associations of the SNPs with the disease, resulting in the inflation of their observed association effects. Therefore, we consider the SNPs used before the selection step to address the effect size inflation issue. Here we define the pre-selected SNPs as the SNPs used in building the PRS model before running any selection related to association strengths. For C+T, the pre-selected SNPs are genome-wide SNPs collected in the training GWAS data. In this situation, we have $\omega_m = 0$ for SNPs not selected for building PRS in C+T. Therefore, we opt for a Gibbs-sampling-based SAME algorithm in C+T to consider the correlations across genome-wide SNPs. In contrast, LD pruning is a selection step unrelated to SNP associations with the disease. Therefore, the pre-selected SNPs in P+T are the SNPs selected after an LD pruning step, and we can use an EM algorithm to derive the distribution of effects in terms of changing allele frequencies between two groups.

We have added this clarification in Section 2.1, Lines 152-167, Pages 4-5.

Q6: In Figure 4, there appears to be a high consistency between PRStuning and testing data results for P+T. Yet, the correlation ρ_{AUC} seems low (0.383-0.783) as shown in Table 7. Conversely, LDpred results seem less consistent in Figure 4, but exhibit a high correlation ρ_{AUC} (0.969-0.987) in Table 7. Could you clarify this discrepancy?

Response: Thank you for your insightful observation. First, we would like to emphasize the complementary roles of ρ_{AUC} and $r_{d_{AUC}}$. We define ρ_{AUC} as the correlation of the PRStuning-

predicted AUC values and those estimated on the testing data, which measures how much the AUC patterns across parameter values for PRStuning and testing data align with each other. Different from ρ_{AUC} , we define rd_{AUC} as the relative difference between the predicted AUC with the best-performing parameter tuned by PRStuning and the AUC with the best-performing parameters on the testing data. In other words, rd_{AUC} measures the point difference between the highest AUC values for the two methods. These two measures reflect different aspects of consistency between PRStuning-predicted AUC and AUC estimated on the testing data for a PRS method.

In Figure 4, we do observe some underestimation of AUC for LDpred (and C+T on CAD and IBD) due to the discrepancy between the reported total sample size and the actual sample size used for computing the z-scores of each SNP (We provide detailed explanation for this issue in Section 2.3, Lines 497-509, Page 18). The underestimation leads to the high values of rd_{AUC} and contributes to the "inconsistency" observed from the figure. However, as we can observe from this figure, the general trends of AUC across different parameters are aligned between the PRStuning and testing data for LDpred. The well-aligned trends lead to the high value of ρ_{AUC} in LDpred.

For P+T, due to the approximate independency among pre-selected SNPs, we do not need to consider the influence of LD in PRStuning and can use an EM algorithm to infer the distribution of effects in terms of changing allele frequencies. The issue of the sample size discrepancy across different SNPs has no influence on the calculation of the covariances among the observed allele frequencies (Second term in Eq (3) for computing τ_j^2), because the covariance term has already been fixed to zero. Therefore, the sample size discrepancy issue is alleviated, leading to the low values of rd_{AUC} and the consistency of AUC observed in Figure 4. However, also note that the standard deviations among the AUC values with different parameters were less than 0.01 for P+T. The extremely small standard deviations of AUC contribute to the large variation of the correlation. Therefore, the correlation is relatively uninformative in characterizing the relationship between the predicted and the actual AUC values.

We provide the explanation for the discrepancy between ρ_{AUC} and the visually consistency of AUC in Section 2.3, Lines 522-530, Page 18.

Q7: Including additional real data results could help further validate the accuracy of PRStuning. For instance, the Breast Cancer Association Consortium has released GWAS summary statistics. Including breast cancer in your analyses as an additional example could be beneficial.

Response: Thank you for your suggestion. We have applied PRStuning on the GWAS summary statistics from the Breast Cancer Association Consortium and used UKBB data to evaluate the performance of PRStuning. The ρ_{AUC} of P+T, C+T, LDpred and LDpred2 are 0.936, 0.956, 0.955 and 0.922, respectively, and the rd_{AUC} of these PRS methods are 1.2%, 0.9%, 0.1% and 0.2%, respectively. The predicted AUC for different PRS methods under different parameter values can be found in Figure 4. The low values of rd_{AUC} indicate that the prediction performance under the PRStuning-selected parameter approximated the best performance on the testing data

accurately. The high values of ρ_{AUC} indicate that PRStuning can accurately predict the pattern of AUC with respect to the parameters on the testing data.

Q8: Traditional PRS analyses often divide data into three independent sets: training, testing, and validation. The prediction could still be overfitted toward the testing data if only the best performance on the testing dataset is reported. While PRStuning can estimate the AUC for the testing dataset based solely on the GWAS summary statistics of the training dataset, how do you avoid potential overfitting when reporting the best prediction PRS after evaluating all the tuning grids?

***Response:** The basic assumption of PRStuning is that the training and testing datasets are homogeneous, indicating that both datasets come from the same population and therefore share the same LD matrix and expected allele frequencies among controls and cases. The same assumption is also needed for traditional PRS analyses based on an independent validation dataset to tune parameters. If the validation and testing datasets are heterogeneous, the AUC estimated from the validation dataset and the parameter selected based on the estimated results are not accurate. Without additional information about the heterogeneity between these two datasets, it will be challenging to estimate AUC and tune parameters based on training or validation datasets. In PRStuning, we focus on dealing with the overfitting issue when the homogeneous assumption is valid. The adjustment to the selected parameter value based on additional information of the heterogeneity is left as future work. Supplementary Figures 2-12 present the performance of PRStuning when the pooled allele frequency, effect size and LD matrix are different between training and testing datasets. The figures demonstrated that the PRStuning can estimate the AUC well when heterogeneity exists in the pooled allele frequency and LD matrix. However, if heterogeneity between training and testing datasets exists in the effects of changing allele frequencies between controls and cases, the AUC from PRStuning will be overestimated and unreliable.*

We have added this clarification in Section 3, Line 600-622, Page 21.

Q9: Recent PRS literature suggests that instead of selecting the best performing PRS based on the testing dataset, combining all PRSs under a tuning grid using ensemble approaches generally yields superior prediction performance (as illustrated in the following studies:

<https://pubmed.ncbi.nlm.nih.gov/31761295/>

<https://www.biorxiv.org/content/10.1101/2022.03.24.485519v5.full.pdf>

<https://www.biorxiv.org/content/10.1101/2023.03.15.532652v1.full.pdf>

<https://www.biorxiv.org/content/10.1101/2023.04.12.536510v1.full.pdf>

Can PRStuning incorporate an ensemble step into its framework? If not, a discussion on this topic would be useful.

***Response:** Thank you for your suggestion. We agree that combining all PRSs under a tuning grid using ensemble methods can improve the prediction performance [3-6]. In the ensemble*

methods, an independent validation dataset is needed to estimate the weights used for combining PRS. In PRStuning, we estimate the AUC and select the best-performed parameters for a PRS method based on the SNP weights derived from the PRS method. If the PRS weights used in ensemble methods have already been estimated in an individual-level validation dataset, we can combine the SNP weights in each PRS and the PRS weights together to derive the ensembled SNP weights. In this situation, PRStuning can be used to predict the AUC of the PRS from the ensembled weights without another individual-level dataset. However, without individual-level validation dataset to estimate the PRS weights used in the ensemble methods, PRStuning can not estimate the PRS weights simply based on GWAS summary statistics from the training data.

We have added this discussion in Section 3, Lines 623-636, Pages 21-22.

Q10. This paper presents convincing evidence, through extensive simulations and real data analyses, that the proposed PRStuning approach can produce consistent results with an external testing dataset. However, additional theoretical intuition would be valuable. Specifically, it's somewhat counterintuitive why incorporating shrinkage via the Bayesian framework can entirely eliminate potential overfitting.

Response: Thank you for your suggestion. In Supplementary Note Section 5, we provide a theoretical demonstration of how overfitting happens if we simply plug the summary statistics from the training data into the AUC function. The rationale of alleviating overfitting with the Bayes estimator is also provided in the same section.

References

1. Wang, Y., Namba, S., Lopera, E., Kerminen, S., Tsoo, K., Läll, K., Kanai, M., Zhou, W., Wu, K.H., Favé, M.J. and Bhatta, L., 2023. Global Biobank analyses provide lessons for developing polygenic risk scores across diverse cohorts. *Cell Genomics*, 3(1).
2. Pain, O., Glanville, K.P., Hagenaars, S.P., Selzam, S., Fürtjes, A.E., Gaspar, H.A., Coleman, J.R., Rimfeld, K., Breen, G., Plomin, R. and Folkersen, L., 2021. Evaluation of polygenic prediction methodology within a reference-standardized framework. *PLoS Genetics*, 17(5), p.e1009021.
3. Privé, F., Vilhjálmsdóttir, B.J., Aschard, H. and Blum, M.G., 2019. Making the most of clumping and thresholding for polygenic scores. *The American Journal of Human Genetics*, 105(6), pp.1213-1221.
4. Zhang, H., Zhan, J., Jin, J., Zhang, J., Lu, W., Zhao, R., Ahearn, T.U., Yu, Z., O'Connell, J., Jiang, Y. and Chen, T., 2022. Novel methods for multi-ancestry polygenic prediction and their evaluations in 5.1 million individuals of diverse ancestry. *bioRxiv*, pp.2022-03.
5. Zhang, J., Zhan, J., Jin, J., Ma, C., Zhao, R., Connell, J.O., Jiang, Y., 23andMe Research Team, Koelsch, B.L., Zhang, H. and Chatterjee, N., 2023. An Ensemble Penalized Regression Method for Multi-ancestry Polygenic Risk Prediction. *bioRxiv*, pp.2023-03.
6. Jin, J., Zhan, J., Zhang, J., Zhao, R., O'Connell, J., Jiang, Y., 23andMe Research Team, Buyske, S., Gignoux, C.R., Haiman, C.A. and Kenny, E., 2023. ME-Bayes SL: Enhanced

Bayesian Polygenic Risk Prediction Leveraging Information across Multiple Ancestry Groups. *bioRxiv*, pp.2023-04.

Reviewer #3:

Jiang et al. proposed PRStuning to automatically tune hyperparameters for PRS calculation using summary statistics from the training data. There are many statistical and machine learning methods have been developed to calculate PRS, but all of them require independent validation sets with raw genotype data to finetune the hyperparameter. I believe PRStuning is timely and would draw much attention in the field. Generally speaking, the manuscript is well prepared and with clear structures, but it lacks evidence to prove it outperforms the other methods.

Response: We appreciate the reviewer for summarizing the contributions of our work. Please find our point-by-point responses below for addressing your concerns. We include more evidence in the manuscript to prove the good performance of our method. Hope these responses are satisfactory.

Major:

1. The authors should discuss and benchmark the methods solving the same problem in the manuscript, such as PUMAS <https://doi.org/10.1186/s13059-021-02479-9>. It is not clear why PRStuning outperforms PUMAS which also uses summary statistics for model selection.

Response: Thank you for your suggestion. We compared PRStuning with PUMAS [1], a method to estimate predictive R^2 for PRS models using summary statistics from GWAS by sampling pseudo-summary-statistics. To compare predictive R^2 with AUC, we first converted Pearson's correlation to Spearman's rank correlation and then linearly mapped the latter to AUC [2]. In Supplementary Table 1, we summarize ρ_{AUC} and $r_{d_{AUC}}$ based on PUMAS. We observe that PRStuning outperformed PUMAS across all real data and PRS methods, and that PUMAS is especially incapable of predicting the AUC well for LDpred and LDpred2.

We have added this discussion in Section 2.3, Lines 533-540, Page 18.

2. Regarding the manuscript, it is observed that PRStuning has only been tested on the UK Biobank dataset. The manuscript does not clarify the performance of PRStuning in cross-population studies and how to address differences in LD patterns. The authors should address the following two scenarios: 1) The summary statistics used for training data are from a multi-ancestry meta-analysis and the testing data is from different populations, and 2) The summary statistics used for training data are from Caucasian populations and the testing data is from Asia, Africa, etc.

Response: The basic assumption of PRStuning is that the training and testing datasets are homogeneous, indicating that both datasets come from the same population and therefore share the same LD matrix and expected allele frequencies among controls and cases. In PRStuning, we currently focus on dealing with the overfitting issue when the homogeneous assumption is valid, meaning that both datasets come from the same population and therefore share the same LD matrix and expected allele frequencies among controls and cases. Therefore, PRStuning may have unreliable performance in the multi-ancestry and heterogenous scenarios

suggested by the reviewer. To investigate which heterogenous factors contribute to the unreliable performance, Supplementary Figures 3-13 present the performance of PRStuning when the pooled allele frequency, effect size and LD matrix are different between training and testing datasets. The figures demonstrate that the PRStuning can estimate the AUC well when the heterogeneity exist in the pooled allele frequency and LD matrix. However, if the heterogeneity between training and testing datasets exists in the effects of changing allele frequencies between controls and cases, the AUC from PRStuning will be overestimated and unreliable.

We have added the descriptions in Section 3, Lines 600-622, Page 21.

3. Follow-up the last question, PRStuning should also be applied to the methods specifically designed for cross-population research, such as PRS-CSx, TL-PRS, SDPRX.

Response: We note that some recent PRS methods have been proposed to consider multiple populations from different ancestries together, which can transfer the knowledge from European population to other demographics with limited sample size [3-6]. As we described in the response of Point 2, we currently focus on dealing with the overfitting issue when the homogeneous assumption is valid, meaning that both datasets come from the same population and therefore share the same LD matrix and expected allele frequencies among controls and cases. We do not consider PRS methods specifically designed for cross-population research in homogeneous scenario. We agree that the adjustment to the selected parameter value based on additional information of the heterogeneity is important and will be addressed in our future work.

4. In Figure 1, please interpret why the difference between unadjusted AUC and PRStuning is much larger when lenient p-value thresholds are applied. Does it mean more inflation would be observed by involving more SNPs?

Response: From the definition of AUC in Eq.(2), we know that AUC is monotonically increasing with respect to Δ . We have $\Delta \propto \zeta := \sum_{m=1}^M \omega_m \delta_m$, which is directly influenced by the SNP effects on the disease. In Supplementary Note Section 5, we derived the difference in P+T between the expectation of ζ and the expectation of $\hat{\zeta}_{train} := \sum_{m=1}^M \omega_m s_m z_m$, where δ_m in ζ is replaced by its observed value $s_m z_m$ from training data.

We conclude that

$$E(\hat{\zeta}_{train}) - E(\zeta) = \sum_{m=1}^M s_m^2 g_m(t),$$

where t is the threshold used to filter SNPs based on z-scores to construct PRS, and

$$g_m(t) = t \left(\phi \left(t - \frac{\delta_m}{s_m} \right) + \phi \left(-t - \frac{\delta_m}{s_m} \right) \right) + \left(1 - \Phi \left(t - \frac{\delta_m}{s_m} \right) + \Phi \left(-t - \frac{\delta_m}{s_m} \right) \right).$$

Here ϕ and Φ are probability and cumulative density functions of $N(0,1)$, respectively. Taking derivative of $g_m(t)$, we get

$$g'_m(t) = t \left(\phi' \left(t - \frac{\delta_m}{s_m} \right) - \phi' \left(-t - \frac{\delta_m}{s_m} \right) \right) < 0,$$

which indicates the overfitting phenomenon will be more severe when more SNPs are involved in PRS calculation.

This explains why the difference between unadjusted AUC and PRStuning is much larger when lenient p -value thresholds are applied. Details of our derivation can be found in Supplementary Note Section 5.

5. The authors replicated 50 times in simulating independent SNPs while replicated only 20 times in simulating correlated SNPs. Is there a reason for reducing the number of replicates?

Response: Simulating genotypes of correlated SNPs costs more time than simulating genotypes of independent SNPs (around 10 times faster than simulating correlated SNPs). In the previous version of our manuscript, we presented results of simulating correlated SNPs with 20 replicates due to the computational concern. In the revised manuscript, we updated simulation results of correlated SNPs based on 50 runs (Line 338, Page 10).

6. Adding an additional metric to measure the proportion of shared cases between the best-performing hyperparameter from PRStuning and the testing data would be helpful. This metric would provide more detailed information for comparing PRStuning with the benchmark.

Response: To further demonstrate the effectiveness of PRStuning, we calculate the sensitivity of the PRS model tuned by PRStuning, which is the proportion of true cases among predicted ones from the PRS model. The cutoff value for PRS is selected by Youden's J statistic, which is defined as the sum of sensitivity and specificity minus one and is the most commonly used criterion to select the cutoff value for a binary classifier [7]. The true case proportions of experiments for four PRS methods are summarized in the Supplementary Figure 2, Tables 4-6 and Supplementary Table 2.

7. Please justify the criteria for determining the number of cases and controls in the simulation.

Response: In simulation experiments, we fixed the case-control-ratio (CCR) to one. This is a common scenario when the GWAS data is collected from a retrospective design in practice [8]. With CCR=1, the actual sample size of the GWAS and the effective sample size $n_{eff} = \frac{4n_0n_1}{n_0+n_1}$ are equal. We can investigate the performance of PRStuning with respect to different effective sample sizes by increasing the total sample size of the GWAS.

8. Can the authors show more details of $\delta = SRS^{(-1)}\beta$ in the section 4.3?

Response: Thank you for your question and sorry for the confusion. In the manuscript, $\delta_m = f_{1,m} - f_{0,m}$ is defined as the difference between the marginalized allele frequencies in the case and control groups for SNP m . We use $p_{0,m}$ and $p_{1,m}$ to denote the potential allele frequencies in

the two groups under an assumed condition that SNP m is independent with other SNPs. We further define $\beta_m = p_{1,m} - p_{0,m}$ as the underlying effect of SNP m in terms of changing allele frequencies between the two groups. In the Supplementary Note Section 1, we proved that $\delta = (\delta_1, \dots, \delta_M)$ is actually related to the LD among the pre-selected SNPs and $\beta = (\beta_1, \dots, \beta_M)$:

$$\delta = SRS^{-1}\beta.$$

Details of the derivation can be found in the Supplementary Note.

9. Song et al. (PMID: 30911754) pointed out that SummaryAUC was not suitable for PRS models including all common SNPs, e.g. LDpred. As PRStuning is built on the top of SummaryAUC and this fact was not fully discussed in the manuscript.

Response: Song et al. pointed out that SummaryAUC was not suitable for PRS models including all common SNPs in the genome due to the reason of accuracy and computational efficiency. In SummaryAUC, the marginal allele frequency differences observed from summary statistics are directly plugged into the calculation of AUC. As we proved in Section 5 of Supplementary Note, if the summary statistics are derived from the training data, overfitting will occur, and the estimated AUC will have upward bias. Even if the summary statistics are derived from another independent validation data, SummaryAUC ignores the correlation between the marginal allele frequency differences due to LD, leading to the inaccurate estimation of AUC when correlated common SNPs are included in the PRS models.

In contrast, we improved the accuracy of estimating the AUC in the following two aspects:

- 1) Instead of using observed allele frequency difference in the AUC function, we estimate allele frequency difference based on the Empirical Bayes theory. The shrinkage effect of the Bayes estimator can alleviate the influence of overfitting. Details can be found in Supplementary Note Section 5.
- 2) The influence of LD on the marginal allele frequency difference, i.e., $\delta = SRS^{-1}\beta$, is also considered in PRStuning.

To reduce the computational burden, we use LDetect to divide the whole genome into approximately independent blocks [9]. For human genomes with European ancestry, a total of 1,703 blocks are partitioned by LDetect.

10. Can the authors provide all required files of an example to run PRStuning on Github?

Response: Thank you for your suggestion. We have updated our Github page (at <https://github.com/lscientific/PRStuning>) and uploaded an example data to run PRStuning.

Minor:

1. In line 74, should it be 'and four normal distributions' rather than 'and three normal distributions'?

Response: Thank you for pointing out this typo. We have changed the sentence to “sBayesR performs Bayesian posterior inference based on a mixture prior of point and three normal distributions that represent SNPs with small, medium, and large effects respectively.”

2. It seems that ‘parameters’ and ‘hyperparameters’ are used interchangeably in the manuscript, which should be avoided. For example, in line 88, ‘evaluate different parameter values’ is supposed to be ‘evaluate different hyperparameter values’. In addition, in line 91, ‘just for tuning parameters’ is supposed to be ‘just for tuning hyperparameters’.

Response: Thank you for your suggestion. We have replaced ‘hyperparameters’ with ‘parameters’ in order to be consistent with the title.

3. In line 416, ‘PRS methods have proven useful’ -> ‘PRS methods have been proven useful’.

Response: Thank you for pointing out this typo. We have made the modifications accordingly in the revised manuscript.

4. In line 474, ‘For example, the frequency’ -> ‘For example, the frequencies’.

Response: Thank you for pointing out this typo. We have made the modifications accordingly in the revised manuscript.

5. When testing the codes on Github, I found that I was required to install rpy2 package and R, but this was not pointed out in the dependency part of Github. Such information should be included.

Response: Thank you for trying out our package. We have updated our Github page (<https://github.com/lscientific/PRStuning>) and added all required packages in the dependency section.

6. In the dependency part of Github, ‘pysnptools scikit-learn Pandas arspy’ is not separated by ‘;’.

Response: We have updated our Github page and separated each package by ‘;’.

7. The data and code availability section should be added to the manuscript.

Response: We have added the data and code availability section in the manuscript.

References

1. Zhao, Z., Yi, Y., Song, J., Wu, Y., Zhong, X., Lin, Y., Hohman, T.J., Fletcher, J. and Lu, Q., 2021. PUMAS: fine-tuning polygenic risk scores with GWAS summary statistics. *Genome Biology*, 22, pp.1-19.

2. Gneiting, T. and Walz, E.M., 2022. Receiver operating characteristic (ROC) movies, universal ROC (UROC) curves, and coefficient of predictive ability (CPA). *Machine Learning*, 111(8), pp.2769-2797.
3. Cai, M., Xiao, J., Zhang, S., Wan, X., Zhao, H., Chen, G. and Yang, C., 2021. A unified framework for cross-population trait prediction by leveraging the genetic correlation of polygenic traits. *The American Journal of Human Genetics*, 108(4), pp.632-655.
4. Ruan, Y., Lin, Y.F., Feng, Y.C.A., Chen, C.Y., Lam, M., Guo, Z., He, L., Sawa, A., Martin, A.R. and Qin, S., 2022. Improving polygenic prediction in ancestrally diverse populations. *Nature Genetics*, 54(5), pp.573-580.
5. Zhao, Z., Fritsche, L.G., Smith, J.A., Mukherjee, B. and Lee, S., 2022. The construction of cross-population polygenic risk scores using transfer learning. *The American Journal of Human Genetics*, 109(11), pp.1998-2008.
6. Zhou, G., Chen, T. and Zhao, H., 2023. SDPRX: A statistical method for cross-population prediction of complex traits. *The American Journal of Human Genetics*, 110(1), pp.13-22.
7. Bantis, L.E., Nakas, C.T. and Reiser, B., 2014. Construction of confidence regions in the ROC space after the estimation of the optimal Youden index-based cut-off point. *Biometrics*, 70(1), pp.212-223.
8. Kraft, P. and Cox, D.G., 2008. Study designs for genome-wide association studies. *Advances in Genetics*, 60, pp.465-504.
9. Berisa, T. and Pickrell, J.K., 2016. Approximately independent linkage disequilibrium blocks in human populations. *Bioinformatics*, 32(2), p.283.

REVIEWER COMMENTS

Reviewer #1 (Remarks to the Author):

Report: Tuning Parameters for Polygenic Risk Score Methods Using GWAS Summary Statistics from Training Data

The revised manuscript addressed most of my comments.

It would be highly appreciated if Authors could also clarify the following points:

- Authors mentioned that the software will accept only PLINK binary format (BED file) as input to calculate LD matrix. While converting the dosage to hard calls there could be loss of information. Would it be an issue if the user wants to use their own imputed data sets to calculate LD matrices?
- The description of Figure 4 is not clear, it would be great to update it with the four techniques included in the Figure. Instead of Figure, if the AUCs are given in a table which will be more readable than figure.
- As the LDpred2 had convergence issues when the risk SNP proportion was set to an extremely small value, the authors have chosen values as {1, 6e-1, 3e-1, 1e-1, 6e-2, 3e-2, 1e-2}. Does it mean that PRStunning with LDpred2 won't work on highly polygenic traits with possible smaller number of causal variants? Could Authors please clarify that whether the Authors prefer to use PRStunning with LDpred over LDpred2?

Reviewer #2 (Remarks to the Author):

The authors have fully addressed my comments in their response.

Reviewer #3 (Remarks to the Author):

All my concerns have been addressed.

Tuning Parameters for Polygenic Risk Score Methods Using GWAS Summary Statistics from Training Data

Reviewer #1:

The revised manuscript addressed most of my comments. It would be highly appreciated if Authors could also clarify the following points:

1. Authors mentioned that the software will accept only PLINK binary format (BED file) as input to calculate LD matrix. While converting the dosage to hard calls there could be loss of information. Would it be an issue if the user wants to use their own imputed data sets to calculate LD matrices?

Response: Thank you for your suggestion. We agree that converting dosage to hard calls may lose information in the downstream analyses. Therefore, we have improved our software so that users can also input Oxford-format binary genotype file (.bgen) to calculate the LD matrix. In our software, we can extract the dosage information from the .bgen file and calculate the LD matrix based on it. The updated software can be downloaded from <https://github.com/lscientific/PRStuning>.

2. The description of Figure 4 is not clear, it would be great to update it with the four techniques included in the Figure. Instead of Figure, if the AUCs are given in a table which will be more readable than figure.

Response: Thank you for your suggestion, and sorry for the confusion made in the description of Figure 4. We have already revised the description to align with the four adopted PRS techniques in the figure. We highlighted the changes in the red color. Besides, we have provided the detailed AUC information of the four diseases for each method and each parameter in Supplementary Table 1.

3. As the LDpred2 had convergence issues when the risk SNP proportion was set to an extremely small value, the authors have chosen values as {1, 6e-1, 3e-1, 1e-1, 6e-2, 3e-2, 1e-2}. Does it mean that PRStunning with LDpred2 won't work on highly polygenic traits with possible smaller number of causal variants? Could Authors please clarify that whether the Authors prefer to use PRStunning with LDpred over LDpred2?

Response: Sorry for the confusion. The convergence issue of LDpred2 simply means that LDpred2 is unable to generate PRS weights in our synthetic experiments when the proportion of causal variants is set to a value smaller than 1e-2. In other words, PRS can only be built with LDpred2 when the parameter is above 1e-2, thus we selected the parameter from {1, 6e-1, 3e-1, 1e-1, 6e-2, 3e-2, 1e-2}. This range is reasonable since the actual proportion of causal variants used to simulate phenotype was set to 5e-2, and PRS is expected to work well when the parameter is close to the actual value.

In our experiments, PRStuning with LDpred had ρ_{AUC} above 0.876 and rd_{AUC} smaller than 2.9% (Table 3). PRStuning with LDpred2 had ρ_{AUC} above 0.881 and rd_{AUC} smaller than 2.5% (Supplementary Figure 1). Note that a high value of ρ_{AUC} indicates that the predicted AUC using PRStuning is highly correlated with the AUC on the testing data, and a small value of rd_{AUC} indicates that the tuning parameter selected by PRStuning and the actual best-performing parameter have comparable performances. The results of the two metrics for PRStuning with LDpred and LDpred2 demonstrated that the AUC patterns across parameter values for PRStuning and testing data aligned with each other, and the point difference between their highest AUC values was small, in both LDpred and LDpred2. Therefore, combining results from real data experiments, we believe that PRStuning works well for the four PRS methods with tuning parameters, including P+T, C+T, LDpred and LDpred2.

Reviewer #2:

The authors have fully addressed my comments in their response.

Response: Thank you and we really appreciate the invaluable comments and suggestions to our method.

Reviewer #3:

All my concerns have been addressed.

Response: Thank you and we appreciate your comprehensive comments and suggestions.

REVIEWERS' COMMENTS

Reviewer #1 (Remarks to the Author):

The revised manuscript addressed all my comments. Thank you for adding the details to the manuscript.